



# Constraining a data-driven $CO_2$ flux model by ecosystem and atmospheric observations using atmospheric transport

Samuel Upton[1,4], Markus Reichstein[1], Wouter Peters[4,5], Santiago Botía[2], Jacob A. Nelson[1], Sophia Walther[1], Martin Jung[1], Fabian Gans[1], László Haszpra[6,7], and Ana Bastos[1,3]

[1]Department of Biogeochemical Integration, Max Plank Institute of Biogeochemistry, Jena Germany
[2]Department of Biogeochemical Signals, Max Plank Institute of Biogeochemistry, Jena Germany
[3]Institute for Earth System Science and Remote Sensing, Leipzig University
[4]Environmental Sciences Group, Wageningen University, Wageningen, The Netherlands
[5]University of Groningen, Centre for Isotope Research, Groningen, The Netherlands
[6]International Radiocarbon AMS Competence and Training Center, HUN-REN Institute for Nuclear Research, Debrecen, Hungary
[7]Institute of Earth Physics and Space Science, Sopron, Hungary

**Correspondence:** Samuel Upton (supton@bgc-jena.mpg.de)

**Abstract.**

Global estimates of the terrestrial land-atmosphere flux of $CO_2$ (NEE) from data-driven models differ widely depending on their underlying data and methodology. Bottom-up models trained on eddy-covariance data are most informative at the ecosystem-level. Top-down models, such as atmospheric inversions, produce regional and global results consistent with the

observed atmospheric growth rate, accurately capturing the interannual variability (IAV) of NEE. Both approaches have limitations estimating NEE across scales: Bottom-up models can miss large-scale dynamics of NEE when aggregated globally. Top-down approaches have difficulty relating the large-scale atmospheric signal to biophysical processes at smaller scales. To address these limitations, we create a model that uses a hybrid combination of direct observations and atmospheric dynamics to integrate ecosystem-level eddy-covariance data and atmospheric $CO_2$ mole fraction data into a single coherent ecosystem-level

flux model.

Aggregated globally, our new model estimates an annual sink with a low bias, and consistent IAV when compared with independent estimates. The IAV of the estimated NEE is closer in magnitude to an ensemble of atmospheric inversions, and our model produces a higher temporal coefficient of correlation with these data than state-of-the-art bottom-up data-driven models. This improvement in IAV is achieved without direct access to the observed variability of the atmosphere: the model is

trained using only one year of daytime observations from 3 tall-tower observatories. No atmospheric information is available to the model during the production of global NEE estimates. This shows the efficiency of our method in synthesizing top-down information into bottom-up mapping of flux-environment relationships.





## 1   Introduction

The global net flux of biogenic $CO_2$ between the land surface and the atmosphere, or net ecosystem exchange (NEE), is a
critical but uncertain term in our understanding of the global carbon budget and the climate system (Friedlingstein et al., 2014;
Bastos et al., 2022). Current data-driven approaches to modeling NEE can be largely grouped into two categories: top-down
approaches that infer NEE from observations of atmospheric $CO_2$ by prescribing fire and fossil fuel emissions. The second
category are bottom-up approaches that model NEE at the ecosystem level from local observations (Kondo et al., 2020). Both
approaches provide important information about the function of the biosphere, but have their limitations.

Data-driven bottom-up models are most commonly informed by eddy-covariance observations at the ecosystem scale, along
with remotely sensed variables (greenness and water indices, spectral data) (Nelson* and Walther* et al., 2024; Jung et al.,
2020; Kondo et al., 2020; Bodesheim et al., 2018). However, the spatial distribution of NEE and the total magnitude of NEE
at global scale remain highly uncertain as the available eddy-covariance data is sparse in important biomes across the globe,
like the tropical rain forests (Chu et al., 2017; Hayek et al., 2018; Fu et al., 2018; Jung et al., 2020). Some processes, such as
carbon emissions from fires, are not captured by these local observations. Eddy-covariance data are further potentially subject
to systematic errors overestimating the carbon uptake (Aubinet et al., 2005), particularly in the tropics due to complex $CO_2$
nighttime storage and horizontal advection in the canopy (Fu et al., 2018; Moncrieff et al., 1996). Historically, when these
eddy-covariance data are used to train a data-driven bottom-up flux model for global upscaling of NEE (e.g. FLUXCOM V1,
Jung et al. (2020)), the underlying issues of data availability and representation are propagated into the global result resulting in
an overestimation of the tropical carbon sink and by extension, the global carbon sink. Therefore data-driven global products
such as FLUXCOM V1 NEE, which are very dependent on the quality and completeness of the training data, are not fully
consistent with the magnitude of the growth rate of atmospheric $CO_2$, or the interannual variability (IAV) of the global NEE
signal (Jung et al., 2020; Kondo et al., 2015). However the length of the available eddy-covariance record has increased, and
the data collection and processing have improved (Pastorello et al., 2020). These longer, improved data, along with improved
handling and gap-filling of the complementary co-located remotely sensed variables (Walther* and Besnard* et al., 2022)
have allowed for FLUXCOM X-BASE to improve its estimates of the magnitude of regional and global NEE. Despite these
improvements, the eddy-covariance record may miss complex drivers of IAV, or they may be obscured by sensor issues (Jung
et al., 2024). The under-estimation of the global NEE IAV in FLUXCOM X-BASE remains unresolved (Nelson* and Walther*
et al., 2024).

Top-down approaches, most commonly atmospheric inversions, are trained using observations of $CO_2$ mole fraction from
tall-tower observatories and/or satellite retrievals of $CO_2$. Atmospheric inversions use a Bayesian inversion framework and
an atmospheric transport model to produce global or regional estimates of NEE which are consistent with the observed at-
mospheric signal (Chevallier et al., 2005; Peylin et al., 2013; Crisp et al., 2022; Ciais et al., 2022). These inversion systems
by design provide estimates of NEE that agree with the growth rate of $CO_2$ at the global scale and across broad latitudinal
bands (Peylin et al., 2013) despite differences in priors and representation of transport mechanisms. Atmospheric inversions





are formulated to capture the structure and dynamics of atmospheric $CO_2$, and can reproduce the observed IAV of NEE with very high accuracy (Rödenbeck et al., 2018).

However, despite improvements in the ability of regional inversions to provide spatially explicit flux estimates (Munassar et al., 2022), global top-down estimates of NEE lack the ability to spatially map the atmospheric signal to local biophysical conditions. Global inversions are more commonly used to understand the land surface at larger integrated scales, and are not directly comparable with eddy-covariance data (Kaminski and Heimann, 2001). Like bottom-up systems, the top-down approach is also limited by a lack of observations. The observational network does not provide sufficient tropical coverage for a robust top-down estimate of the tropical land flux (Palmer et al., 2019). In the southern hemisphere, the weaker gradients in $CO_2$, and shorter observational records also exacerbate issues of data availability and data quality (Peylin et al., 2013). Additionally, in tropical regions there are less independent data to validate atmospheric inversion estimates (Chevallier et al., 2019).

In previous work it was demonstrated that top-down information can be effectively combined with a bottom-up data-driven flux model, improving the regional and global performance by creating a dual constraint from eddy-covariance data and atmospheric inversion estimates of NEE (Upton et al., 2024). Although this dual-constrained model was able to infer regional and global NEE integrals with a much lower bias compared with other best estimates of NEE than the comparable FLUXCOM RS+METEO V1 results (Jung et al., 2020), the model has several important limitations; First, the atmospheric constraints were pre-calculated using large-scale mean NEE from an ensemble of inversions, rather than from atmospheric $CO_2$ itself. Second, the atmospheric constraint was connected to the local bottom-up data-driven flux model using a static, statistical model rather than an atmospheric transport model. This means that the additional atmospheric information is aggregated, and adds no additional information on the spatial distribution of NEE. Both issues limited the amount of information available to the model from the atmospheric constraint.

To address these limitations, we create a new model that uses a hybrid combination of direct observations and atmospheric dynamics to integrate ecosystem-level eddy-covariance data and atmospheric $CO_2$ mole fraction data into a single coherent ecosystem-level flux model. To create the computational link between ecosystem-level fluxes and atmospheric $CO_2$ mole fraction we use a physical atmospheric transport model, the Stochastic Time-Inverse Lagrangian Transport (STILT) model (Lin et al., 2003). In this way, both eddy-covariance and atmospheric observations are used to train and update the model at the ecosystem level, the result is a bottom-up model that encodes top-down information. This technique opens a large corpus of new observations for constraining the model. Due to varying transport patterns around each tower, the data-driven model is exposed to a large number of non-eddy-covariance locations, enriching the model's view of the underlying relation between driver variables and NEE. These changes, we hypothesize, allow the system to learn additional dynamics in global, and regional NEE.





## 2  Data

### 2.1  Ecosystem-level data

At the ecosystem level, we define Net Ecosystem Exchange (NEE) as the simple difference between Gross Primary Production

(GPP) and Total Ecosystem Respiration (TER). These terms are measured as the flux density $\mu mol CO_2\ m^{-2}\ s^{-1}$. The sign of NEE indicates the direction of the flux. When NEE is negative, GPP exceeds TER and the flux is a local sink of $CO_2$. If NEE is positive, TER exceeds GPP and the ecosystem is a local source. NEE represents the net carbon exchange of the vegetation and does not include disturbance fluxes, fires, or out-gassing from freshwater ecosystems.

The core ecosystem-level meteorological and NEE data was collected between 2001 and 2020 at 294 globally distributed

eddy-covariance (EC) towers (see appendix for a full list of sites). These data were then processed using the ONEFLUX processing pipeline (Pastorello et al., 2020). Variables from the eddy-covariance sites are air temperature, vapor pressure deficit, incoming shortwave radiation, the computed potential shortwave incoming radiation and the computed wind speed.

During training, at pixels co-located with eddy-covariance towers we use the surface reflectances (MCD43A4, Schaaf and Wang (2015a)) and land surface temperature (LST) from the MOderate Resolution Imaging Spectroradiometer (MODIS)

collection v006 (http://daac.ornl.gov/MODIS/). The surface reflectances are used to compute two vegetation indices, the enhanced vegetation index (EVI) (Huete et al., 2002), and the spectral reflectance of vegetation in the near-infrared (NIRv) (Badgley et al., 2019). A water index, the normalized difference water index (NDWI) with MODIS band 7 (Gao, 1996) was also included. MODIS data has a spatial resolution of 0.05 degrees, and a 1-day temporal resolution

When producing our atmospheric constraint, and when estimating global NEE, the drivers for our bottom-up model are the

same set of remotely-sensed and eddy-covariance quantities, extracted from remotely-sensed and meteorological reanalysis data. The RS data are derived from the 0.05 degree global MODIS product (MCD43C4, Schaaf and Wang (2015b)), and the global meteorological variables are derived from the European Center for Medium-Range Weather Forecasting (ERA5) atmospheric reanalysis data (https://www.ecmwf.int/en/forecasts/dataset/ecmwf-reanalysis-v5). ERA5 is provided at 0.25 degree spatial resolution, and hourly temporal resolution (Hersbach et al., 2020). Air temperature at 2m height, incoming shortwave

radiation, and vapor pressure deficit (computed from air temperature, relative humidity, and surface pressure) are used to create the global meteorological data.

All data are accessed through the FLUXCOM-X code base using the preprocessing, and gap-filling from Nelson* and Walther* et al. (2024) and the procedures of FluxnetEO data version 2 (Walther* and Besnard* et al., 2022), and the quality flagging described in Jung et al. (2023). Please see Nelson* and Walther* et al. (2024) for a full description of the FLUXCOM-

X data and processing environment.

### 2.2  Atmospheric data

The model in this study uses observations of atmospheric $CO_2$ mole fraction from tall-tower observatories measured at three sites: ATTO (Botia et al., 2022), Hegyhatsal (Haszpra, 2024) and Zottino (Tran et al., 2024a). These sites are chosen to include one tropical, one extra-tropical and one boreal domain. The mole fraction data used for the Amazon Tall Tower Observatory





in Brazil (ATTO), collected at 79 meters are the same used and fully described in Botia et al. (2022). Mole fraction data for the Hegyhátsál tower in Hungary (HUN), collected at 115 meters are provided by the Integrated Carbon Observatory System (ICOS). The observations for the Zotino Tall Tower Observatory in Siberia (ZOTTO), collected at 301 meters are provided by Tran et al. (2024b). The data for all three towers was provided hourly, for one calendar year (2019 for ZOTTO and Hegyhátsál, 2015 for ATTO). Only daytime observations from 13:00-17:00 (local time) were used. The selected times represent a daytime

planetary boundary layer (PBL) with assumed well-mixed convective conditions (Peters et al., 2010).

## 2.3 Reference data

For global and regional comparison we use the land flux corrected for fossil fuel and cement production from the ensemble of N=14 atmospheric inversions from GCB23 (https://doi.org/10.18160/4M52-VCRU, Luijkx et al. (2024), last accessed 4.3.2024). The atmospheric inversions are provided as monthly data with a 1°spatial resolution. See Friedlingstein et al. (2023)

for a full description of the ensemble members. These fluxes are adjusted for fire by removing the fire emissions from Global Fire Emissions Database, Version 4.1 (GFEDv4) (last access 13.2.24) (RANDERSON et al., 2017).

For comparison with the current state-of-the-art bottom-up NEE model, we use the global NEE product from the latest version of FLUXCOM X-BASE (Nelson* and Walther* et al. (2024), last access January 2025). The FLUXCOM X-BASE NEE model (hereafter X-BASE) is trained using the same data processing and data quality flagging, including gap-filling.

X-BASE uses a different data-driven algorithm, XGBoost (Chen and Guestrin, 2016), instead of a neural network. Comparison with X-BASE allows us to understand the performance of our system relative to a highly optimized bottom-up model. So, despite differences in structure, we consider X-BASE to be the most appropriate bottom-up model for comparison.

For regional analysis we use the set of 11 large regions from the TransCom 3 intercomparison project (TransCom) project (Baker et al., 2006). See appendix figure A1 for the coverage and names of the 11 land regions used in this study.

## 3 Methods

In this work, we start with a neural network model which uses eddy-covariance, meteorological, and remotely sensed data to estimate NEE at the ecosystem level. We create a hybrid objective function which allows direct constraint by eddy-covariance (EC) observations and hybrid constraint from direct mixing ratio observations of $CO_2$ using the Stochastic Time-Inverted Lagrangian Transport (STILT) atmospheric transport model as the computational link from the EC-model to the top-down

observations. We therefore refer to this new data-driven system with its dual constraints as EC-STILT. Its components and their recombination in a machine-learning framework are described below.

### 3.1 Ecosystem-level model

The ecosystem-level model takes as input observations of meteorological drivers (from eddy-covariance or reanalysis data) and remotely-sensed drivers and predicts NEE in $\mu mol CO_2\ m^{-2}\ s^{-1}$ (Fig. 1, red lines) at an hourly tempo. As a machine-learning

system, the ecosystem-level model can be described as a feed-forward neural network trained using standard backpropagation





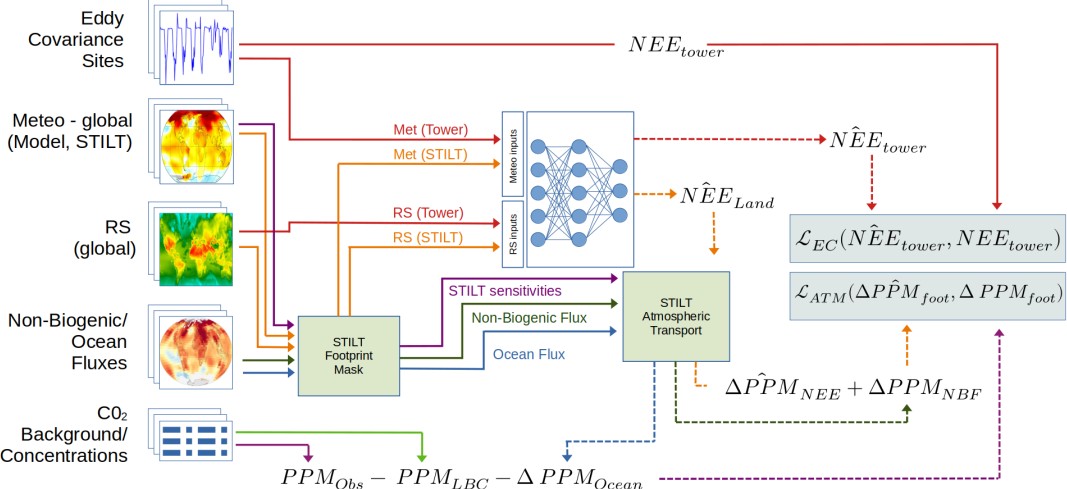

**Figure 1.** Data flow for atmospheric constraint calculation. An objective term from eddy-covariance (red lines) is created from tower observations of driver variables, co-located remotely sensed data, and tower observations of NEE. An objective term from atmospheric observations (model - orange lines, ancillary data - other colors) uses the STILT model to transport modeled flux densities into concentration space, and corrected for non-local and non-biogenic $CO_2$: The lateral boundary condition ($PPM_{LBC}$), or background $CO_2$ from the well-mixed atmosphere, and any ocean flux contributions ($\Delta PPM_{Ocean}$) are removed from the tower observation ($PPM_{Obs}$) to create the observed value for the contribution from the footprint ($\Delta PPM_{foot}$). Non-biogenic fluxes ($\Delta PPM_{NBF}$) from fire and fossil fuels are added to the predicted NEE from the footprint ($\Delta P\hat{P}M_{NEE}$) to produce the predicted contribution from the footprint ($\Delta P\hat{P}M_{foot}$). Dashed lines indicate terms that are created during the process of training. See sections 3.3 and 3.4 for a full description of terms.

techniques (Kelley, 1960). The network is a set of fully-connected layers which consist of nodes or "neurons". The neurons are exposed to the output of all neurons in the previous layer. Between each layer of neurons, the output is passed through a non-linear activation function. Additionally, the output of the activation function is passed through a normalization function, which scales the output to a mean of 0 and a standard deviation of 1. Our network is a set of three fully-connected layers with

the Rectified Linear Unit (ReLU) activation function (Agarap, 2019) and batch normalization. Each layer in a neural network is a complex, non-linear embedding of the previous inputs into a new latent space. This series of embedding functions is the learning space of the network. These latent spaces allow the model to learn complex non-linear relationships between driver variables. Because these relationships are discovered in these abstract non-geographic spaces, they can represent complex space-for-time and time-for-space substitutions.

## 3.2 STILT Transport

The computational link from this ecosystem-level model to the atmospheric constraint (Fig. 1, purple lines) is built around the STILT atmospheric transport model (Lin et al., 2003). STILT is a Lagrangian particle dispersion model (LPDM) which uses meteorological forcing data to compute the transport of sources and sinks to the time and location of an observation. The



STILT footprint represents this atmospheric transport as a set of sensitivities of the final observation to fluxes at different times
and locations. This footprint can be multiplied with a set of local estimates of NEE, which transports them and converts them
into a simulated mole fraction. EC-STILT uses a set of pre-computed STILT footprints at the three tall tower locations, for
midday observations, for one year (Fig. 1). For every selected atmospheric mole fraction, for each tower, over the year, the
STILT model is run for the specific tower domain, hourly for 10 days prior to the observation time. STILT domains (Fig. 2)
are selected to capture the relevant near field over the time of analysis based on regional meteorological conditions (Lin et al.,
165  2003).

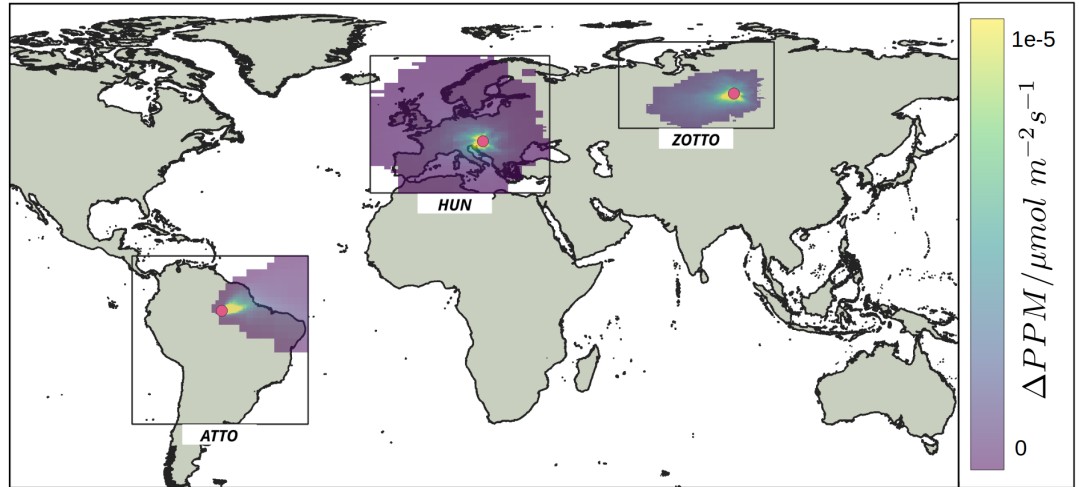

**Figure 2.** Locations of tall towers used by the atmospheric constraint (red dots). The boxes represent the near-field domain where
STILT is computed, and the grid within each domain is the mean annual sensitivities of the computed STILT footprints. The colorbar,
in $\Delta PPM/\mu mol\, m^{-2} s^{-1}$, represents the mean sensitivity of the tower observation to the region over the full year.

For the atmospheric constraint, we must isolate the biogenic contribution ($\Delta PPM_{NEE}$) of an hourly observation of $CO_2$
($PPM_{Obs}$). Conceptually, the biogenic contribution is the residual of the observation and the non-biogenic terms: the lateral
boundary condition (LBC), non-biogenic fluxes (NBF) and Ocean contributions. This is presented in equation 1, with spatially
explicit terms on the right.

$$PPM_{Obs} - PPM_{LBC} - \Delta PPM_{Ocean} = \Delta PPM_{NEE} + \Delta PPM_{NBF} \qquad (1)$$

The NBF contribution ($\Delta PPM_{NBF}$) consists of fossil fuel fluxes and fire emissions. For fossil fuel emissions, we use fossil
fuel/energy signal across sectors, using GridFED (version 2022.2, last access - 15.2.24) (Jones et al., 2021), provided at a 0.1
deg resolution. Fire emissions are provided by the Global Fire Emissions Database, Version 4.1 (GFEDv4) (last access 13.2.24)
(RANDERSON et al., 2017) provided at a 0.25 deg. For the GFEDv4 data, temporal profiles were used to add monthly, daily,
and diurnal cycles to the fire signal. The GridFED data are interpolated to the 0.25 degree spatial resolution of the analysis.



The lateral boundary condition of the region ($PPM_{LBC}$) is precomputed using The Jena CarboScope (s04ocv4.3) (Rödenbeck et al., 2003), following Botia et al. (2022). Optimized results from the CarbonTracker Data Assimilation Shell (CTDAS) (van der Laan-Luijkx et al., 2017) (CTE2020) are used for the ocean flux and its conversion to mole fractions ( ($\Delta PPM_{Ocean}$) with STILT.

**3.3 Objective function - Eddy covariance**

During training, EC-STILT generates an objective function with two terms, or losses: the ecosystem loss term $\mathcal{L}_{EC}$, and the atmospheric loss term $\mathcal{L}_{ATM}$. To generate the ecosystem loss term $\mathcal{L}_{EC}$ from eddy-covariance, the EC-model $\mathcal{F}()$ is run using a batch of data randomly selected by time and site, $x_{\text{batch}}$ collected at eddy-covariance tower sites (Eq. 2, along with remotely sensed (RS) data at pixels that are co-located with the eddy-covariance tower, producing an estimate of NEE at the ecosystem

level (eq. 2). The ecosystem loss term is computed as the mean squared error (MSE) between this estimate and the observed NEE from the eddy-covariance observations (Eq. 3).

$$N\hat{E}E = \mathcal{F}(x_{\text{batch}}) \tag{2}$$

$$\mathcal{L}_{EC} = MSE(NEE_{\text{obs}}, N\hat{E}E) \tag{3}$$

**3.4 Objective function - Atmospheric**

To generate the atmospheric loss term $\mathcal{L}_{ATM}$ for each training step, an hourly daytime (13:00 to 17:00 local time) observation of the mole fraction of $CO_2$ for each tower is selected. For each observation, a pre-computed STILT footprint, the LBC and ocean data are selected to create the left-hand side of eq 1. This is the change in $CO_2$ mole fractions attributable to fluxes within the footprint ($\Delta PPM_{foot}$) that we aim to reproduce with the NEE model. The ecosystem-level model is then run for each non-zero location in the STILT footprint, producing an estimate of local NEE. The NEE inferences and NBF values are transported

with the footprint, and the two terms are then added element-wise. The result is summed to create the simulated change in PPM at the tower which is the right-hand side of eq 1. The loss term $\mathcal{L}_{tower,ATM}$ is the squared difference between the simulated change in concentration $\Delta P\hat{P}M_{foot}$, and the observed change in concentration $\Delta PPM_{foot}$. For a single training step, a loss is calculated for one observation for every tower. These three tower losses are averaged, producing the final atmospheric loss term (Eq. 5).

$$\mathcal{L}_{tower,ATM} = (\Delta PPM_{foot} - \Delta P\hat{P}M_{foot})^2 \tag{4}$$

$$\mathcal{L}_{ATM} = \frac{\sum_t^{Num.Towers} \mathcal{L}_{t,ATM}}{Num.Towers} \tag{5}$$

The two terms of the objective function, $\mathcal{L}_{EC}$ and $\mathcal{L}_{ATM}$ are combined using a learned weighting scheme (Kendall et al., 2018). This allows the model to learn the appropriate relative weight to assign each loss term during training, based on their internally estimated uncertainty. The model has a set of additional parameters which are used to learn an estimate of the

homoscedastic uncertainty for each term of the objective function. This uncertainty is dependent on the inherent noise in the





training data, rather than the quality or quantity of training data. For EC-STILT these mixing parameters, $\sigma^2_{EC}$ and $\sigma^2_{ATM}$, are added to the processing chain of the model after its initialization, and then during training, the normal backpropagation process that neural networks use to update their parameters, can also update these mixing parameters. For the individual tasks $L \in [EC, ATM]$, the $\sigma^2_L$ parameter is used to create two terms; $w_{\mathrm{L}}$ (Eq. 6), and $s_{\mathrm{L}}$ (Eq. 6). These are then used to calculate

the effective loss, balanced by the learned uncertainty of the terms (Eq. 8). After this final weighting calculation, the loss is combined in small batches to stabilize model training during backpropagation.

$$w_{EC} = \frac{1}{2\sigma^2_{EC}}, \quad w_{ATM} = \frac{1}{2\sigma^2_{ATM}} \tag{6}$$

$$s_{EC} = \log\sqrt{\sigma^2_{EC}}, \quad s_{ATM} = \log\sqrt{\sigma^2_{ATM}} \tag{7}$$

$$\mathcal{L}_{total} = (w_{EC} \times \mathcal{L}_{EC}) + (w_{ATM} \times \mathcal{L}_{ATM}) + s_{EC} + s_{ATM} \tag{8}$$

## 3.5 Model Training

EC-STILT is trained using 10-fold cross-validation. The full set of eddy-covariance observations are split by tower location into 10 equal subsets. Ten ensemble members are created, each trained on 8 of the subsets, with one held out for validation and one held out for testing. The ensemble is used to define the flux uncertainty as given in the Results sections, as a 1-$\sigma$ of the member spread. The available footprints for each tower were randomized and 80% are used for training, 20% for validation on

each member. The cross-validation results for the eddy-covariance level results below are the ensemble member results against the 10% of sites held out for testing.

The STILT footprints were run for a 0.25 degree grid. All MODIS driver variables and NBF data were interpolated to this grid. MODIS variables were interpolated bilinearly. NBF fluxes were interpolated using the nearest neighbor method which reduced the overestimation of emissions in pixels surrounding urban hotspots. When run globally, EC-STILT is run at full

MODIS 0.05 degree resolution.

# 4 Results

## 4.1 Global model performance

EC-STILT produces an estimate of annual global NEE of -3.90 $PgC\ year^{-1} \pm 3.63$, compared to the ensemble mean of atmospheric inversions in GCB23, adjusted for fire, -4.29 $PgC\ year^{-1} \pm 0.65$ (Fig. 3 B). The RMSE of the EC-STILT annual

mean with the inversion ensemble annual mean is 0.69 $PgC\ year^{-1}$, compared with 1.81 $PgC\ year^{-1}$ for X-BASE. The uncertainty of the global annual flux is very high (Fig. 3 A), being driven by uncertainties in tropical NEE (Fig. 4C).

EC-STILT has learned a relationship between the input variables and NEE, which captures a large part of the natural inter-annual variability of global annual NEE. When the model is run globally for years 2001-2021, with only local driver variables, and no atmospheric information, the IAV of the EC-STILT member mean, 0.69 PgC year$^{-1}$ (1-$\sigma$), is substantially closer to




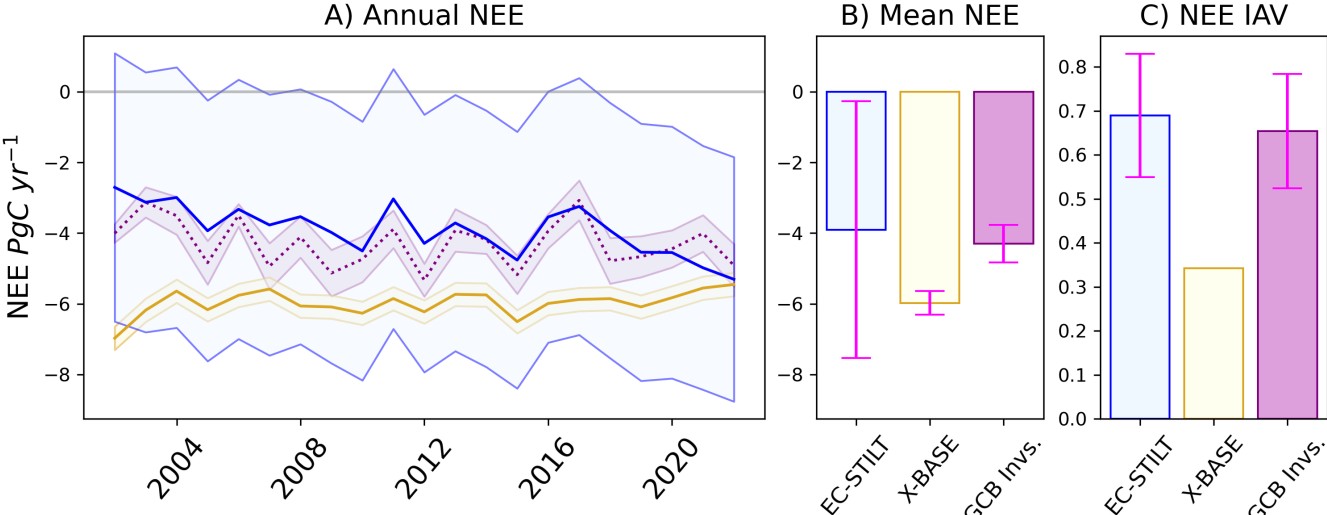

**Figure 3.** A) Annual NEE integrated globally in $PgC\,year^{-1}$. The blue line is the mean of the EC-STILT ensemble with the shaded area representing the 1-$\sigma$ range across ensemble members. The yellow line is FLUXCOM X-BASE. The purple dotted line is the mean of the GCB23 ensemble of atmospheric inversions, and the shaded area is the standard deviation across the ensemble members. B) Mean NEE integrated globally over all years. The error bars represent the 1-$\sigma$ range across ensemble members or published uncertainty. C)IAV of NEE, calculated as the standard deviation of the global integral. The error bars represent the 1-$\sigma$ range of IAV across ensemble members

the GCB23 inversion ensemble IAV of 0.65 PgC year$^{-1}$ than that of X-BASE (Fig. 3 C). The R$^2$ of EC-STILT annual mean NEE with the GCB23 inversion ensemble is 0.42 (N=21 years), compared with an R$^2$ of 0.02 for X-BASE. EC-STILT IAV is consistent across the model members despite the large uncertainty in the magnitude of the flux. The mean $R^2$ between the 10 members and GCB23 inversion ensemble is 0.34 (1-$\sigma$ 0.08), and the mean IAV is 0.71 (1-$\sigma$ 0.14) PgC y$^{-1}$.

     Spatially, the distribution of the mean annual NEE is largely consistent with X-BASE (Figs. 4 B) with a spatial correlation of
0.71 for global mean NEE. EC-STILT has a stronger Amazonian sink and weaker boreal sink (Fig. 4, B). The boreal reduction is a reduction in the length and intensity of the growing season. The EC-STILT results have removed several hotspots of source, in the Sahel and in the Indian subcontinent. These potentially unrealistic hotspots in X-BASE can be attributed to an incorrect learned relationship for crop-cover PFTs in certain dry conditions (Nelson* and Walther* et al., 2024). The lack of specific Plant Functional Type (PFT) information in the EC-STILT training data may be responsible for the removal of these strong
sources. The large model spread in the tropics (Fig. 3 C) across the Amazon Basin, the Congo Basin, and Oceania is the major source of global model spread in Fig. 3A.

## 4.2   Regional model performance

The atmospheric constraint has a strong impact on the EC-STILT estimate of regional fluxes. Similar to the global results, EC-STILT estimates regional magnitudes of NEE which are largely inconsistent (Fig. 5, Tab. 1) when compared with the ensemble



**Figure 4.** Spatial distribution of global annual NEE A) Mean annual NEE for EC-STILT in $gC\,m^{-2}\,day^{-1}$ B) The difference mean annual NEE between EC-STILT and X-BASE in $gC\,m^{-2}\,day^{-1}$. C) The standard deviation of mean annual NEE for the EC-STILT 10-member ensemble in $gC\,m^{-2}\,day^{-1}$

of atmospheric inversions used in Friedlingstein et al. (2023) corrected for fire using GFED4.1 fire emissions (Fig. 5, purple lines). The regional IAV is more in agreement with atmospheric inversions, producing similar or better $R^2$ and magnitude when compared with X-BASE regional NEE (Fig. 5, inset text, Tab. 1).

     Regionally, there appears to be a relationship between the coverage of the eddy-covariance and atmospheric observational networks, and the impact of the atmospheric information (Tab. 1). For example, in Europe (Fig. 5), both the atmospheric and

eddy-covariance networks are extensive. It appears that in limited cases, the atmospheric information provides some additional information on the magnitude. EC-STILT is closer to the inversion mean with an RMSE of 0.2 $PgC\,y^{-1}$ compared with 0.37 $PgC\,y^{-1}$ for X-BASE. However, the IAV is very strongly conditioned on the information in eddy-covariance data; the IAV for Europe from EC-STILT is still very similar to X-BASE with a $R^2$ of 0.86 compared with X-BASE, and 0.31 compared with the inversion mean (Tab. 1). The spatial pattern of summer NEE during the peak growing season (mean JJA) is very consistent



**Figure 5.** Annual regional NEE and IAV by TransCom region. Column A: Regional fluxes for TransCom regions directly impacted by atmospheric observations: Europe is impacted by observations in the HUN domain. Eurasia Boreal is impacted by observations in the ZOTTO domain. Both South American regions are directly impacted by observations in the ATTO domain. (left): the time series of the annual regional integral flux. (right) the magnitude of IAV for the region. Column B: Regions of similar biome and latitudinal band, by row. The blue is the ensemble mean of EC-STILT, the yellow is FLUXCOM X-BASE, the purple is the ensemble mean of the atmospheric inversions in GCB23 with GFED4.1 fire emissions removed. For each annual time series, the $R^2$ is reported for EC-STILT and X-BASE with regard to the fire-corrected ensemble mean

between EC-STILT and X-BASE (Fig. 6) with a spatial correlation of 0.86, demonstrating the strong influence eddy-covariance information retains when used in dual-constraint. The reduced sink in the region estimated by EC-STILT (Fig. 5), appears in Figure 6 to be largely an overall bias, rather than a learned difference in the spatial distribution of NEE.





**Table 1.** Comparison of different estimates with GCB: Column A shows the RMSE between the annual regional integrals in PgC year$^{-1}$ of EC-STILT and X-BASE with the atmospheric inversion ensemble mean with fire removed (Invs). Column B shows the $R^2$ between the annual regional integrals from EC-STILT and X-BASE with regard to the atmospheric inversion ensemble mean, and between EC-STILT and X-BASE. Column C shows RMSE of the monthly IAV between EC-STILT and X-Base with regard to the atmospheric inversion ensemble mean. Bold indicates the higher relative performance between EC-STILT and X-BASE for each region and metric.

| | A) Annual RMSE | | B) Annual $R^2$ | | | C) IAV RMSE | |
| --- | --- | --- | --- | --- | --- | --- | --- |
| | EC-STILT, Invs | X-BASE, Invs | EC-STILT Invs | X-BASE, Invs | EC-STILT, X-BASE | EC-STILT, Invs-f | X-BASE, Invs-f |
| North American Boreal | 1.076 | **0.094** | **0.009** | 0.008 | 0.387 | 0.012 | 0.012 |
| North American Temperate | **0.205** | 0.579 | **0.158** | 0.067 | 0.554 | **0.006** | 0.008 |
| South American Tropical | 3.725 | **2.030** | **0.098** | 0.039 | 0.176 | 0.011 | 0.011 |
| South American Temperate | 0.735 | **0.585** | **0.317** | 0.310 | 0.424 | **0.007** | 0.010 |
| Northern Africa | 1.125 | **0.659** | 0.248 | **0.354** | 0.050 | **0.014** | 0.015 |
| Southern Africa | **0.202** | 0.419 | **0.450** | 0.094 | 0.010 | **0.012** | 0.014 |
| Eurasia Boreal | 0.520 | **0.267** | **0.413** | 0.272 | 0.750 | **0.011** | 0.012 |
| Eurasia Temperate | 1.893 | **1.555** | 0.266 | **0.292** | 0.876 | **0.006** | 0.007 |
| Tropical Asia | 0.941 | **0.511** | **0.263** | 0.124 | 0.007 | 0.009 | 0.009 |
| Australia | **1.374** | 0.418 | 0.083 | **0.292** | 0.009 | 0.005 | 0.005 |
| Europe | **0.210** | 0.397 | 0.306 | **0.375** | 0.873 | 0.008 | **0.007** |

In regions such as South American Tropical where there is limited eddy-covariance coverage to adequately constrain the model, additional atmospheric information does not improve the modeled NEE, and increases the model uncertainty. EC-
STILT has moved away from the inversion mean, compared with X-BASE (RMSE of 3.73 $PgC \ y^{-1}$ compared with 2.05 $PgC \ y^{-1}$). The low $R^2$ for EC-STILT compared with both the inversion mean and X-BASE annual time series (0.10 compared with the inversions, 0.13 compared with X-BASE) indicate that neither the eddy-covariance data nor the atmospheric signal provide sufficient constraint for EC-STILT to find similar IAV to the atmospheric inversion mean, which itself might be wrong.

The increase in the IAV for the South American Temperate region demonstrates the potential for complex effects from the atmospheric constraint. The region is only directly impacted by the ATTO tower observations in unusual meteorological conditions, which is evident in the mean ATTO footprint in Figure 2. The eddy-covariance record for the region is sparse, but may have similarities to other, more densely sampled regions. We see a small move away from the inversion mean compared with FLUXCOM X-BASE in the magnitude of NEE (RMSE of 0.74 $PgC \ y^{-1}$ compared with 0.59 $PgC \ y^{-1}$ for X-BASE), but an increase in the magnitude of IAV compared with FLUXCOM X-BASE (0.14 $PgC \ y^{-1}$ compared with 0.07 $PgC \ y^{-1}$). The atmospheric constraint has added new information about this region despite very limited direct information available in training.

The atmospheric constraint also has strong effects in TransCom regions that do not directly contain or overlap the STILT regions used during training. This is because EC-STILT learns its land-surface response in environmental space of the features instead of in geographic space like an inversion, Across the globe, EC-STILT produces consistent changes over regions with



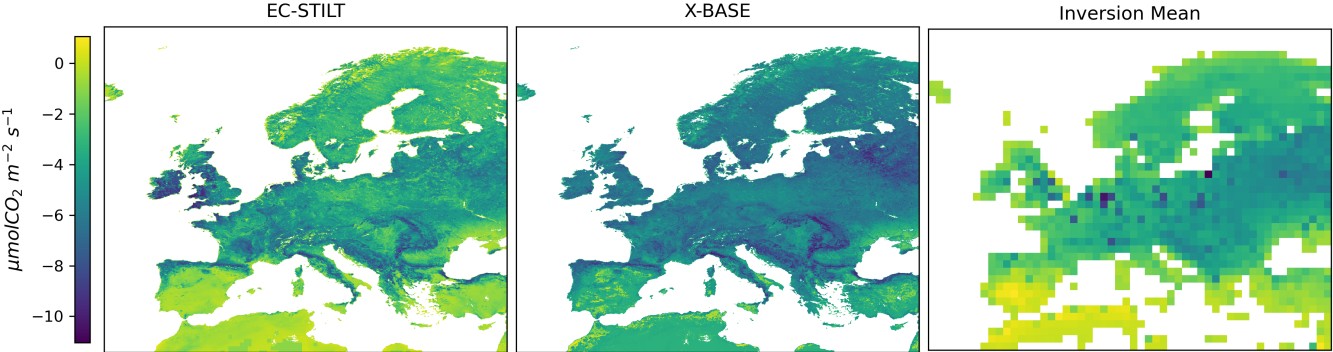

**Figure 6.** Mean summer NEE (JJA) in $\mu mol CO_2 \ m^{-2} \ s^{-1}$ for EC-STILT, X-BASE, and the ensemble mean of the atmospheric inversions.

similar latitude ranges (Fig. 5, right column) with a dominance of the global north in observational networks (Chu et al., 2017). The magnitude of the northern hemispheric land carbon sink is part of ongoing discrepancies in the Global Carbon Budget (O'Sullivan et al., 2024).

### 4.3 IAV attribution

While the global IAV of EC-STILT is closer to IAV of the GCB land residual (Fig 3), the regional impact of atmospheric
constraint is more heterogeneous. The relationship between EC-STILT and the inversion mean appears to depend on the biome, and the strength of the eddy-covariance constraint. In certain regions EC-STILT has improved the $R^2$ (Southern Africa), or the magnitude of IAV (South American Temperate) relative to the fire-corrected ensemble mean of the atmospheric inversions, when compared to X-BASE (Fig. 5). In regions which are well constrained by eddy-covariance (Europe, North American Temperate), both the magnitude and annual $R^2$ relative the inversion mean are very similar to X-BASE. In these regions, the
annual $R^2$ relative to X-BASE is higher than with the inversion ensemble (Tab. 1).

EC-STILT mean monthly IAV, shows very little change compared with X-BASE (Tab. 1, IAV RMSE). EC-STILT has modified its response by biome; When IAV is broken down by month (Fig. 7) across boreal and temperate regions, the increases in monthly IAV occurs during the growing season while in tropical regions show a change during the dry season. Appendix Figure B1 shows the monthly spatial distribution of IAV. The monthly IAV also shows a clear dependence on the growing
season, with IAV distributed from the timing of the onset and end of season, and the magnitude of the carbon uptake. Tropical regions generally have lower IAV (Fig. 7), and EC-STILT locates the more of the IAV in the dry season.

### 4.4 Eddy-covariance site-level evaluation

EC-STILT maintains good overall performance for NEE at the eddy-covariance site level, particularly at daily time scale. We use $R^2$ and the Nash Sutcliffe model efficiency (NSE) (Nash and Sutcliffe, 1970) to evaluate model performance with regard







**Figure 7.** Attribution of IAV across the mean seasonal cycle: The left column is the monthly mean seasonality of IAV for the region, calculated as the standard deviation of the month across the period of analysis (2001-2021). The blue line is the member mean of EC-STILT, the yellow line is the X-BASE NEE, and the purple dotted line is the ensemble mean of the atmospheric inversions in GCB23. The right column is the difference (EC-STILT - X-BASE) across the year.

to the observed eddy-covariance observations, and for comparison with X-BASE. In the EC-STILT's 10-fold cross-validation for all sites, the model achieves a median NSE of 0.66, and median $R^2$ of 0.77 for hourly fluxes (X-BASE: NSE 0.71, $R^2$ 0.71), and a median NSE of 0.76 and median $R^2$ of 0.87 for diurnal mean NEE (X-BASE: NSE 0.81, $R^2$ 0.81), when comparing the observed and simulated time series. The data used for these performance metrics are the test data, held out from each cross-validation fold, and these tests are performed using the model member which did not see these data in training.





As with regional results, the site-level performance of EC-STILT appears to be strongly influenced by the level of local constraint from eddy-covariance. In figure 8 the hourly and daily averaged time series of NEE for three EC sites directly impacted by the atmospheric domains, one in boreal Russia (RU-Ha1), one in a German mixed forest (DE-Hai), and one Brazilian site in the Amazon basin (BR-Sa1) are shown. In the Russian site, EC-STILT (blue line) matches the temporal pattern of the EC observations (red line) at the hourly level ($R^2$ of 0.57). When averaged to the daily scale, both EC-STILT and

X-BASE produce a similar RMSE compared with the observations. EC-STILT appears to better represent the daily signal than X-BASE with a higher $R^2$ (0.43 compared with 0.18). EC-STILT produces a range in NEE which is slightly larger than the EC record and much greater than FLUXCOM X-BASE (yellow line). Boreal regions are only moderately constrained in EC-STILT by eddy-covariance given the large difference in annual NEE between EC-STILT and X-BASE (Fig. 5). This may allow for larger differences at the site level, as the boreal observations do not sufficiently weigh against the atmospheric information.

In the German site, DE-Hai, where the constraint from eddy-covariance is strongest, EC-STILT underestimates the hourly magnitude (Fig. 8, center left) with a mean RMSE of 0.92 for EC-STILT, 0.22 for X-BASE), but still largely captures the daily NEE, but with a larger bias than X-BASE (Fig 8, center right) with an $R^2$ of 0.61 and RMSE of 2.00 compared with an $R^2$ of 0.83 and RMSE of 1.04 for X-BASE. In the northern extra-tropics, as seen above, the marginal impact of the atmospheric information is minimized, and EC-STILT performs most similarly with X-BASE. EC-STILT has a stronger positive bias than

X-BASE (EC-STILT 1.75, X-BASE 0.83). This bias in the site level results is consistent with the slightly weaker regional sink seen in the northern temperate regions.

    In the Brazilian site, at the hourly scale EC-STILT strongly underestimates the range of NEE (hourly RMSE of 6.32), but infers a daily value when averaged, that is comparable with X-BASE ($R^2$ of 0.47 for EC-STILT and X-BASE). This may indicate that the eddy-covariance and atmospheric constraints are providing confounding information about the hourly flux, which is discussed below.


## 5   Discussion

The core logic of our model is a synergistic sharing of information between the bottom-up and top-down constraints. The key innovation in our system is the combination of the atmospheric term with the ecosystem-level objective term derived from eddy-covariance data, and their joint projection onto a space that is not the "traditional" geographic distribution of NEE.

Instead, EC-STILT learns in "environmental response" space, or a set of relations between the driver datasets (features) and the output NEE. The STILT Atmospheric transport model provides the spatiotemporal locations where ecosystem NEE variations contribute to a downwind $CO_2$ observation. This large number of locations over the near-field can be modeled and their output constrained by the atmospheric observation. Considered independently this is underconstrained; one atmospheric observation is insufficient to constrain the multiple modeled NEE estimates under the STILT footprint as also experienced in many inverse

modeling problems using atmospheric data (see Peylin et al. (2013); Gaubert et al. (2023) for an overview). However, because the 'environmental response' space of the model is conditioned by the ecosystem-level objective term, the response across the STILT footprint is tightly bound to the local, better-constrained eddy-covariance record. Learning is thus jointly informed by





**Figure 8.** EC tower results across latitude bands: Each row is a tower. The left column is the mean hourly NEE, the right column is the daily mean across an example month. The blue line is the member mean of EC-STILT, the yellow line is the X-BASE NEE, and the red dotted line is the eddy-covariance observations for the tower. All values are in $\mu molC\ m^{-2}\ s^{-1}$.

both data sources, while maintaining, outside of the tropics, the local daily performance of bottom-up models like FLUXCOM V1 (Jung et al., 2020) and X-BASE (Nelson* and Walther* et al., 2024).



EC-STILT efficiently learns driver/NEE relationships globally which adds new information on the regional and global dynamics of NEE. This is demonstrated by the improved representation of IAV of global NEE (Fig. 3) and regional NEE (Fig. 5). There is no explicit representation in the training data of the atmospheric IAV; The atmospheric constraint is only present for three regional domains (around the ZOTTO, ATTO, and HUN towers). These domains were chosen to represent a tropical (ATTO), extra-tropical (HUN), and boreal (ZOTTO) eco-region. Each tower has observations across a single year (2019 for

ZOTTO and Hegyhátsál, 2015 for ATTO) chosen for reasons of data availability. These relatively small samples of the available atmospheric information are sufficient to embed the atmospheric signal in the ecosystem response of EC-STILT, and modify the spatial and temporal distributions of estimated NEE. There is no explicit representation in the data of the atmospheric growth rate, nor mass-balance yet these are reproduced to an impressive degree. The atmospheric observations are regional and instant, with no direct information on the long-term state of the atmosphere. During training, the LBC signal does pro-

vide some indirect information on the atmosphere, as it is removed from the observation. Because the atmospheric transport system is in fact not even run during the calculation of global NEE. There is therefore no atmospheric information available to EC-STILT during the creation of our global NEE data. The potentially realistic global integral and spatial distribution (Fig. 3) demonstrates that the model is able to learn a more consistent ecosystem-level response globally from this sparse set of local observations. Of particular note is that these limited observations improve the estimate of IAV across the entire period of

analysis, and thus learned relations hold under a range of environmental conditions in each pixel.

    Although subject to large uncertainty, the magnitude of the annual flux has moved closer to the GCB23 atmospheric inversion estimate of global NEE. The global magnitude is strongly conditioned, however on the inclusion and handling of the non-biospheric flux data during training; As the total amount of NBF in a region increases, by necessity, the inferred NEE must decrease to keep the local instantaneous equation balanced (Eq. 1). The total contributions to the well-mixed atmosphere over

time are complex (Ciais et al., 2022), and often not available in spatially explicit data products. The current study used fire and fossil fuel emissions, but other terms such as riverine fluxes (see Botía et al. (2024) for a discussion at the ATTO tower) may play strong local or global roles in the regional $CO_2$ balance, and should be included in the tower loss calculation.

    In figure 3A, it appears that the atmospheric signal increases the uncertainty around the magnitude and distribution of tropical NEE, when compared with X-BASE. The ensemble members show a large spread across tropical regions, and at the

ecosystem-level, the ensemble mean has difficulty reproducing the magnitude of hourly observations from the eddy-covariance record (Fig. 8). EC-STILT's tropical shortcomings may be a combination of several structural factors; model type, potential mismatches in the atmospheric operator, and limited observational support, which are discussed below.

    When compared with X-BASE, the core model structure is important; X-BASE is a XGBoost regressor, which have been demonstrated to be robust in this domain compared with neural networks which have more capacity to overfit (Nelson* and

Walther* et al., 2024; Kraft et al., 2024; Jung et al., 2020). Because our objective function is sensitive to ecosystem and regional results across all three towers, it is possible that within the overall learning process, some local or regional overfitting is occurring. We attempt to reduce overfitting using early stopping of training according to the validation accuracy, and in the model using batch normalization. Potentially, this overfitting could lead to large intra-model uncertainty between neural network members compared with an XGBoost ensemble. Current implementations of XGBoost in the Python programming





language (compatible with the FLUXCOM-X framework) were not considered, because they do not allow for complex objective functions with multiple terms, sensitive to different spatial scales. Any additional experimentation with different model structures was outside of the scope of this study.

The tropical response from EC-STILT is also confounded by a specific effect in the performance of the STILT atmospheric operator during training. For HUN and ZOTTO, EC-STILT achieved high Pearson's R values (0.65-0.8) for the inferred and observed land contribution to the $CO_2$. However for ATTO, although the RMSE during training between the inferred and observed land contribution was similar to the training RMSE for HUN and ZOTTO, Pearson's R was near zero (0.0 - 0.15). This means that the system was able to infer the correct magnitude of the contribution, but unable to get the signal correctly located in time. This poor temporal registration may be attributed to a missing NBF contribution to the signal which has a strong temporal component, such as the mentioned riverine evasion of $CO_2$ (Botía et al., 2024). Or the issue may be with the transport model or its meteorological forcing. Our STILT footprints are calculated at 0.25 degree spatial resolution. This resolution was chosen because of computational constraints in our training. It is possible that this spatial resolution does not adequately capture some aspect of the atmospheric processes in the tropics. In relation to the relatively small contribution of intra-model uncertainty within LPDMs (Hegarty et al., 2013), meteorological uncertainty is a more important term in the overall transport uncertainty (Angevine et al., 2014). However, it was out of scope for the current work to estimate that uncertainty with regard to the output STILT sensitivities and and resulting concentration estimates. Lastly, the temporal issue at ATTO may have to do with the rate of air exchange through the canopy. This could introduce a time lag into the release of $CO_2$ out of the canopy and into the free atmosphere (Faassen et al., 2024).

Both bottom-up (Chu et al., 2017) and top-down (Palmer et al., 2019) observational records are limited and uncertain in the tropics. This limited record forces a data-driven model to extrapolate its response for a range of missing or under-sampled parts of the tropical distribution of the driver variables. It is unrealistic for a model constrained only by the existing observational record to move meaningfully beyond previous work on estimating tropical fluxes. Although the boreal region around ZOTTO is also subject to the same uncertainties with regard to the observational record, the uncertainty relative to the magnitude of the flux does not impact global results to the same extent.

In the eddy-covariance site results (Fig. 8) we may see an example of atmospheric information confounding and augmenting the eddy-covariance signal. Both boreal and tropical regions are less well constrained by eddy-covariance than the northern extra-tropics, which increases the marginal impact of the atmospheric constraint. EC-STILT was able to reproduce both the Russian site's (RU-Ha1) eddy-covariance hourly time series and the ZOTTO time series with high fidelity. Indeed, EC-STILT estimates an hourly variability which is higher than the hourly eddy-covariance data. In the Brazilian site (BR-Npw), where EC-STILT was unable to reproduce the ATTO time series, we see the eddy-covariance finding a low-variability estimate of the daily mean. This may represent confounding between constraints.

## 5.1 Future work

This study demonstrates a novel method for combining bottom-up and top-down information in the objective function of a data-driven flux model. This study finds improvement in a number of important metrics in a bottom-up NEE model that is





additionally constrained by three atmospheric tall-towers over one year. As noted above, neither the eddy-covariance record nor the atmospheric observational record have strongly representative global coverage, and additional atmospheric information over new geographic regions could be a benefit. Therefore an important future development of a dual-constrained model is the inclusion of additional tower and non-tower data streams: Additional tall towers could provide stronger constraint in well-measured regions. Non-tower data could be leveraged to provide information on regions that are poorly covered by the current observational network. The current model could run, as-is, against aircraft observations. With a modification of the STILT transport system in the objective function, the structure of this study would allow a bottom-up model to be constrained by satellite retrievals of total column $CO_2$ ($XCO_2$). $XCO_2$ observations could be selected according to their potential to include biomes of specific interest in their upwind footprints. This would enrich our capacity to model NEE across un- and under-sampled biomes. This extension into $XCO_2$ introduces several possible issues: to model the full column of $CO_2$, multiple STILT footprints must be run for every observation at different elevations (Wu et al., 2018), and modeled fluxes transported multiple times. This may be a technical challenge to build effective training systems. The inclusion of the full column $CO_2$ may dilute the available information on the biogenic flux in the observation, reducing the information gain to the bottom-up model.

The performance of EC-STILT in the tropics show the limitations of adding atmospheric information without adequate support in the eddy-covariance record. In the harmonized model described above, the ecosystem-level information appears to provide the 'backbone' of the model's performance. Future modeling efforts will benefit from the ongoing efforts to improve and enlarge the corpus of available eddy-covariance sites and data, such as a potential extension of the FLUXNET 2015 dataset (Pastorello et al., 2020). As the eddy-covariance record better describes and constrains key biomes, additional atmospheric constraint may be able to provide improved information on the larger dynamics of regional and global NEE, while reducing the uncertainty.

An important aspect of transport modeling which is not addressed in the current study is the impact of model and meteoro-logical uncertainty. In our analysis, we treat the output of the STILT model as definitive, relative to the meteorological fields in ECMWF short-term forecast. Computational schemes for addressing model uncertainty, multi-LPDM ensembling, or in-model ensembling for STILT by sampling within the stochastic particles could be attempted, without a major increase in computa-tional and I/O workload. In this study, STILT runs are forced with ECMWF short-term forecast data (Botia et al., 2022), and the model is trained using ERA5 reanalysis data. Using identical meteorological data for STILT and the model drivers may also reduce uncertainty.

Conceptually, the current work accounts for two major terms in the non-biogenic flux budget; fires and fossil fuel emissions. In the Amazon and in boreal Russia, fire is the largest and the most relevant term, as the term accounts for both fire, and also for part of the instantaneous flux from land-cover/land-use change associated with biomass burning (Cochrane and Laurance, 2008). In the European domain, fossil fuels are the dominant non-biogenic flux signal. As noted in the section 2, NEE at the eddy-covariance tower and atmospheric $CO_2$ are not directly comparable. The dual-constraint in this study is created using the residual of an observation and non-biogenic fluxes. Any meaningful non-biogenic flux term which is not included, or adequately represented temporally and spatially will alias into the atmospheric target that the model is trying to match in



training. This reduces the effectiveness of the technique. Any additional spatially explicit NBF data may therefore, improve
this comparison, and increase the fidelity of the overall magnitude and may reduce the uncertainty in the global result.

Beyond an improved global NEE product, a key scientific goal is a detailed understanding of the different learned relationship
in EC-STILT between NEE and the driver variables, and how this mapping is transformed by the inclusion of atmospheric
information. There are many open questions remaining on how EC-STILT locates the IAV signal, and what new spatiotemporal
dependencies are learned outside of the eddy-covariance record. Methods for explainable machine-learning may allow for an
analysis of the model's learned 'environmental space'. Relationships between this internal representation and environmental
variables may provide insight into how and where EC-STILT locates the IAV.

## 6  Conclusions

This study demonstrates a novel method for combining bottom-up and top-down information in the objective function of a
data-driven flux model. We wanted to test the effectiveness of a physics-based operator that transports modeled flux density
into atmospheric concentration. And we wanted to determine whether a temporally and geographically-sparse constraint could
produce valid global results. The EC-STILT model in this study is still strongly conditioned by the bottom-up constraint from
eddy-covariance, and so maintains the strengths of the overall bottom-up approach, but now with access to information on land
surface far beyond the eddy-covariance network.

A classical top-down atmospheric inversion matches the observed atmospheric signal but lacks the framework to accurately
distribute that signal down into geographic/biophysical space. Bottom-up models can translate EC measurements into their
latent space, but do not have any additional information about how inferred fluxes influence the global signal. In contrast, our
hybrid system in this study can take information from the atmospheric observations and accurately distribute it into the learned
latent space of the model. Because this latent space still contains information from the eddy-covariance record, this allows for
a valid mapping down from the atmosphere to the model, and from the model to an estimate of NEE at multiple scales which
is comparable both with both atmospheric and ecosystem-level observations. Using our system, a dual-constrained model sits
directly between bottom-up and top-down approaches, directly addressing their limitations while maintaining their advantages.

## Appendix A: Data

Table A1: Citation data for the 294 sites used in training.

| AR-SLu (Garcia et al., 2016) | AR-TF1 (Kutzbach, 2021) | AR-Vir (Posse et al., 2016) | AT-Neu (Wohlfahrt et al., 2016) | AU-ASM (Cleverly and Eamus, 2016b) | AU-Ade (Beringer and Hutley, 2016c) |
|---|---|---|---|---|---|



Table A1: Citation data for the 294 sites used in training.

| | | | | | |
|---|---|---|---|---|---|
| AU-Cpr (Meyer et al., 2016) | AU-Cum (Pendall and Griebel, 2016) | AU-DaP (Beringer and Hutley, 2016b) | AU-DaS (Beringer and Hutley, 2016f) | AU-Dry (Beringer and Hutley, 2016e) | AU-Emr (Schroder et al., 2016) |
| AU-Fog (Beringer and Hutley, 2016a) | AU-Gin (Macfarlane et al., 2016) | AU-RDF (Beringer and Hutley, 2016d) | AU-Rob (Liddell, 2016) | AU-TTE (Cleverly and Eamus, 2016a) | AU-Tum (Woodgate et al., 2016) |
| AU-Wac (Beringer et al., 2016b) | AU-Whr (Beringer et al., 2016a) | AU-Wom (Arndt et al., 2016) | AU-Ync (Beringer and Walker, 2016) | BE-Bra (Team and Centre, 2022-02-01) | BE-Dor (Team and Centre, 2022-02-01) |
| BE-Lcr (RI, 2021-05-06) | BE-Lon (Team and Centre, 2022-02-01) | BE-Maa (Team and Centre, 2022-02-01) | BE-Vie (Team and Centre, 2022-02-01) | BR-Npw (Vourlitis et al., 2022) | BR-Sa1 (Saleska, 2016) |
| BR-Sa3 (Goulden, 2016d) | CA-Cbo (Staebler, 2022) | CA-DB2 (Knox, 2022) | CA-DBB (Christen and Knox, 2022) | CA-ER1 (Wagner-Riddle, 2021) | CA-Gro (McCaughey, 2016) |
| CA-LP1 (Black, 2021) | CA-Man (Amiro, 2016b) | CA-NS2 (Goulden, 2016a) | CA-NS3 (Goulden, 2016b) | CA-NS4 (Goulden, 2016c) | CA-NS5 (Goulden, 2016g) |
| CA-NS6 (Goulden, 2016e) | CA-NS7 (Goulden, 2016f) | CA-Oas (Black, 2016b) | CA-Obs (Black, 2016a) | CA-Qfo (Margolis, 2016) | CA-SF1 (Amiro, 2016c) |
| CA-SF2 (Amiro, 2016a) | CA-SF3 (Amiro, 2016d) | CA-TP1 (Arain, 2016b) | CA-TP2 (Arain, 2016a) | CA-TP3 (Arain, 2022b) | CA-TP4 (Arain, 2016c) |
| CA-TPD (Arain, 2022a) | CG-Tch (Nouvellon, 2016) | CH-Aws (Team and Centre, 2022-02-01) | CH-Cha (Team and Centre, 2022-02-01) | CH-Dav (Team and Centre, 2022-02-01) | CH-Fru (Team and Centre, 2022-02-01) |
| CH-Lae (Team and Centre, 2022-02-01) | CH-Oe1 (Ammann, 2016) | CH-Oe2 (Team and Centre, 2022-02-01) | CN-Cha (Zhang and Han, 2016) | CN-Cng (Dong, 2016) | CN-Dan (Shi et al., 2016) |
| CN-Din (Zhou and Yan, 2016) | CN-Du2 (Chen, 2016k) | CN-Du3 (Shao, 2016b) | CN-HaM (Tang et al., 2016) | CN-Qia (Wang and Fu, 2016) | CN-Sw2 (Shao, 2016a) |



Table A1: Citation data for the 294 sites used in training.

| | | | | | |
|---|---|---|---|---|---|
| CZ-BK1 (Team and Centre, 2022-02-01) | CZ-BK2 (Sigut et al., 2016) | CZ-KrP (Team and Centre, 2022-02-01) | CZ-Lnz (Team and Centre, 2022-02-01) | CZ-RAJ (Team and Centre, 2022-02-01) | CZ-Stn (Team and Centre, 2022-02-01) |
| CZ-wet (Team and Centre, 2022-02-01) | DE-Akm (Team and Centre, 2022-02-01) | DE-Geb (Team and Centre, 2022-02-01) | DE-Gri (Team and Centre, 2022-02-01) | DE-Hai (Team and Centre, 2022-02-01) | DE-HoH (Team and Centre, 2022-02-01) |
| DE-Hte (Team and Centre, 2020-03-10) | DE-Hzd (Team and Centre, 2022-02-01) | DE-Kli (Team and Centre, 2022-02-01) | DE-Lkb (Lindauer et al., 2016) | DE-Lnf (Knohl et al., 2016) | DE-Obe (Team and Centre, 2022-02-01) |
| DE-RuR (RI, 2022-06-14) | DE-RuS (Team and Centre, 2022-02-01) | DE-RuW (Team and Centre, 2022-02-01) | DE-Seh (Schneider and Schmidt, 2016) | DE-SfN (Klatt et al., 2016) | DE-Spw (Bernhofer et al., 2016) |
| DE-Tha (Team and Centre, 2022-02-01) | DE-Zrk (Sachs et al., 2016) | DK-Eng (Pilegaard and Ibrom, 2016) | DK-Fou (Olesen, 2016) | DK-Gds (RI, 2022-06-14) | DK-Sor (Team and Centre, 2022-02-01) |
| ES-Abr (Team and Centre, 2022-02-01) | ES-Agu (Team and Centre, 2022-02-01) | ES-Amo (Poveda et al., 2016) | ES-Cnd (Team and Centre, 2022-02-01) | ES-LJu (Team and Centre, 2022-02-01) | ES-LM1 (Team and Centre, 2022-02-01) |
| ES-LM2 (Team and Centre, 2022-02-01) | ES-LgS (Reverter et al., 2016b) | ES-Ln2 (Reverter et al., 2016a) | FI-Hyy (Team and Centre, 2022-02-01) | FI-Jok (Lohila et al., 2016) | FI-Ken (Team and Centre, 2022-02-01) |
| FI-Let (Team and Centre, 2022-02-01) | FI-Lom (Aurela et al., 2016a) | FI-Qvd (Team and Centre, 2022-02-01) | FI-Sii (Team and Centre, 2022-02-01) | FI-Sod (Aurela et al., 2016b) | FI-Var (RI, 2022-06-14) |
| FR-Aur (Team and Centre, 2022-02-01) | FR-Bil (Team and Centre, 2022-02-01) | FR-EM2 (RI, 2022-06-14) | FR-FBn (Team and Centre, 2022-02-01) | FR-Fon (Team and Centre, 2022-02-01) | FR-Gri (Team and Centre, 2022-02-01) |
| FR-Hes (Team and Centre, 2022-02-01) | FR-LBr (Berbigier and Loustau, 2016) | FR-LGt (RI, 2022-06-14) | FR-Lam (Team and Centre, 2022-02-01) | FR-Pue (Ourcival, 2016) | FR-Tou (RI, 2022-06-14) |
| GF-Guy (Team and Centre, 2022-02-01) | GH-Ank (Valentini et al., 2016b) | GL-Dsk (RI, 2022-06-14) | GL-NuF (Hansen, 2016) | GL-ZaF (Lund et al., 2016b) | GL-ZaH (Lund et al., 2016a) |



Table A1: Citation data for the 294 sites used in training.

| | | | | | |
|---|---|---|---|---|---|
| IE-Cra (Team and Centre, 2022-02-01) | IL-Yat (Team and Centre, 2022-02-01) | IT-BCi (Team and Centre, 2022-02-01) | IT-BFt (RI, 2022-06-14) | IT-CA1 (Sabbatini et al., 2016c) | IT-CA2 (Sabbatini et al., 2016a) |
| IT-CA3 (Sabbatini et al., 2016b) | IT-Col (Matteucci, 2016) | IT-Cp2 (Team and Centre, 2022-02-01) | IT-Cpz (Valentini et al., 2016a) | IT-Isp (Gruening et al., 2016b) | IT-La2 (Cescatti et al., 2016) |
| IT-Lav (Team and Centre, 2022-02-01) | IT-Lsn (RI, 2022-06-14) | IT-MBo (Team and Centre, 2022-02-01) | IT-Noe (Spano et al., 2016) | IT-PT1 (Manca and Goded, 2016) | IT-Ren (Team and Centre, 2022-02-01) |
| IT-Ro1 (Valentini et al., 2016c) | IT-Ro2 (Papale et al., 2016) | IT-SR2 (Team and Centre, 2022-02-01) | IT-SRo (Gruening et al., 2016a) | IT-Tor (Team and Centre, 2022-02-01) | JP-MBF (Kotani, 2016b) |
| JP-SMF (Kotani, 2016a) | MX-Tes (Yepez and Garatuza, 2021) | MY-PSO (Kosugi and Takanashi, 2016) | NL-Hor (Dolman et al., 2016a) | NL-Loo (Team and Centre, 2020-03-10) | PA-SPn (Wolf et al., 2016b) |
| PA-SPs (Wolf et al., 2016a) | PE-QFR (Griffis and Roman, 2021) | RU-Che (Merbold et al., 2016) | RU-Cok (Dolman et al., 2016b) | RU-Fy2 (Team and Centre, 2022-02-01) | RU-Fyo (Team and Centre, 2022-02-01) |
| RU-Ha1 (Belelli et al., 2016) | SD-Dem (Ardö et al., 2016) | SE-Deg (Team and Centre, 2022-02-01) | SE-Htm (Team and Centre, 2022-02-01) | SE-Lnn (Team and Centre, 2020-03-10) | SE-Nor (Team and Centre, 2022-02-01) |
| SE-Ros (Team and Centre, 2022-02-01) | SE-Svb (Team and Centre, 2022-02-01) | SJ-Adv (Christensen, 2016) | SJ-Blv (Boike et al., 2016) | SN-Dhr (Tagesson et al., 2016) | US-A32 (Billesbach et al., 2022) |
| US-AR1 (Billesbach et al., 2016b) | US-AR2 (Billesbach et al., 2016a) | US-ARM (Biraud et al., 2022) | US-ARb (Torn, 2016b) | US-ARc (Torn, 2016a) | US-Atq (Zona and Oechel, 2016a) |
| US-BZB (Euskirchen, 2022b) | US-BZF (Euskirchen, 2022c) | US-BZS (Euskirchen, 2022d) | US-BZo (Euskirchen, 2022a) | US-Bi1 (Rey-Sanchez et al., 2022b) | US-Bi2 (Rey-Sanchez et al., 2022a) |





Table A1: Citation data for the 294 sites used in training.

| | | | | | |
|---|---|---|---|---|---|
| US-Blo (Goldstein, 2016) | US-CF1 (Huggins, 2021) | US-CF2 (Huggins, 2022c) | US-CF3 (Huggins, 2022a) | US-CF4 (Huggins, 2022b) | US-CRT (Chen and Chu, 2016b) |
| US-CS1 (Desai, 2022a) | US-CS2 (Desai, 2022c) | US-CS3 (Desai, 2022d) | US-CS4 (Desai, 2022b) | US-Cop (Bowling, 2016) | US-EDN (Oikawa, 2021) |
| US-GBT (Massman, 2016) | US-GLE (Massman, 2022) | US-Goo (Meyers, 2016b) | US-HB1 (Forsythe et al., 2021) | US-HWB (Goslee, 2022) | US-Ha1 (Munger, 2016) |
| US-Hn3 (Liu et al., 2022) | US-Ho2 (Hollinger, 2022) | US-IB2 (Matamala, 2016) | US-ICs (Euskirchen et al., 2022a) | US-ICt (Euskirchen et al., 2022b) | US-Ivo (Zona and Oechel, 2016b) |
| US-Jo2 (Vivoni and Perez-Ruiz, 2022) | US-KFS (Brunsell, 2022a) | US-KLS (Brunsell, 2022b) | US-KS1 (Drake and Hinkle, 2016a) | US-KS2 (Drake and Hinkle, 2016b) | US-KS3 (Hinkle, 2022) |
| US-LWW (Meyers, 2016a) | US-Lin (Fares, 2016) | US-Los (Desai, 2016c) | US-MMS (Novick and Phillips, 2022) | US-MOz (Wood and Gu, 2022) | US-Me1 (Law, 2016c) |
| US-Me2 (Law, 2022) | US-Me3 (Law, 2016a) | US-Me4 (Law, 2016e) | US-Me5 (Law, 2016d) | US-Me6 (Law, 2016b) | US-Mpj (Litvak, 2021) |
| US-Myb (Sturtevant et al., 2016) | US-NGB (Torn and Dengel, 2021) | US-NR1 (Blanken et al., 2022) | US-Ne1 (Suyker, 2022) | US-Ne2 (Suyker, 2016b) | US-Ne3 (Suyker, 2016a) |
| US-ONA (Silveira, 2021) | US-ORv (Bohrer, 2021) | US-OWC (Bohrer and Kerns, 2022) | US-Oho (Chen et al., 2016) | US-PFa (Desai, 2016d) | US-Prr (Kobayashi and Suzuki, 2016) |
| US-Rms (Flerchinger, 2022c) | US-Ro1 (Baker et al., 2022) | US-Ro4 (Baker and Griffis, 2022a) | US-Ro5 (Baker and Griffis, 2021) | US-Ro6 (Baker and Griffis, 2022b) | US-Rwe (Flerchinger and Reba, 2022) |
| US-Rwf (Flerchinger, 2022a) | US-Rws (Flerchinger, 2022b) | US-SRC (Kurc, 2022) | US-SRG (Scott, 2016a) | US-SRM (Scott, 2016b) | US-Sne (Shortt et al., 2022) |



Table A1: Citation data for the 294 sites used in training.

| | | | | | |
|---|---|---|---|---|---|
| US-Snf (Kusak et al., 2022) | US-Sta (Ewers and Pendall, 2016) | US-Syv (Desai, 2016b) | US-Ton (Baldocchi and Ma, 2016) | US-Tw1 (Valach et al., 2021) | US-Tw2 (Sturtevant et al., 2022) |
| US-Tw3 (Chamberlain et al., 2022) | US-Tw4 (Sanchez et al., 2016) | US-Tw5 (Valach et al., 2022) | US-Twt (Baldocchi, 2016) | US-UM3 (Bohrer, 2022) | US-UMB (Gough et al., 2016) |
| US-UMd (Gough et al., 2022) | US-Var (Baldocchi et al., 2016) | US-WCr (Desai, 2016a) | US-WPT (Chen and Chu, 2016a) | US-Whs (Scott, 2016d) | US-Wi0 (Chen, 2016g) |
| US-Wi1 (Chen, 2016e) | US-Wi2 (Chen, 2016j) | US-Wi3 (Chen, 2016b) | US-Wi4 (Chen, 2016d) | US-Wi5 (Chen, 2016a) | US-Wi6 (Chen, 2016h) |
| US-Wi7 (Chen, 2016i) | US-Wi8 (Chen, 2016c) | US-Wi9 (Chen, 2016f) | US-Wjs (Litvak, 2022) | US-Wkg (Scott, 2016c) | US-xBR (Network), 2022) |

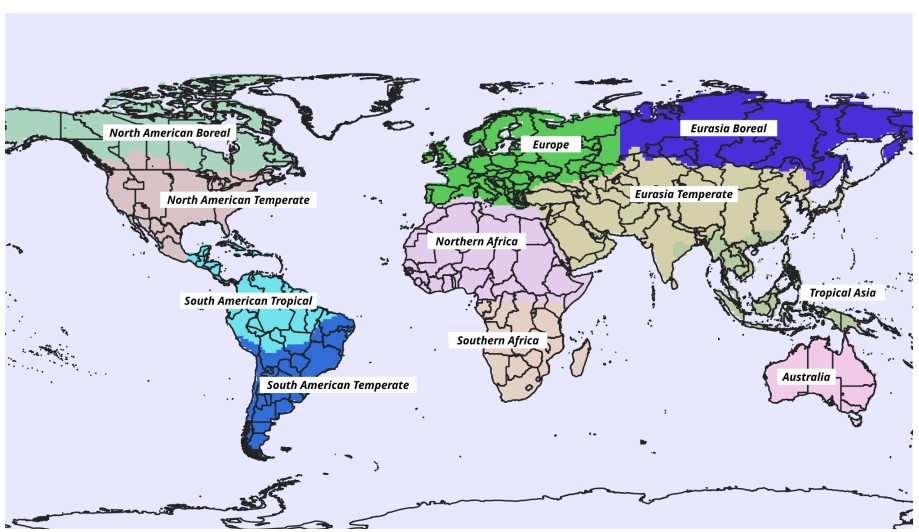

**Figure A1.** TransCom land regions




## Appendix B: Results

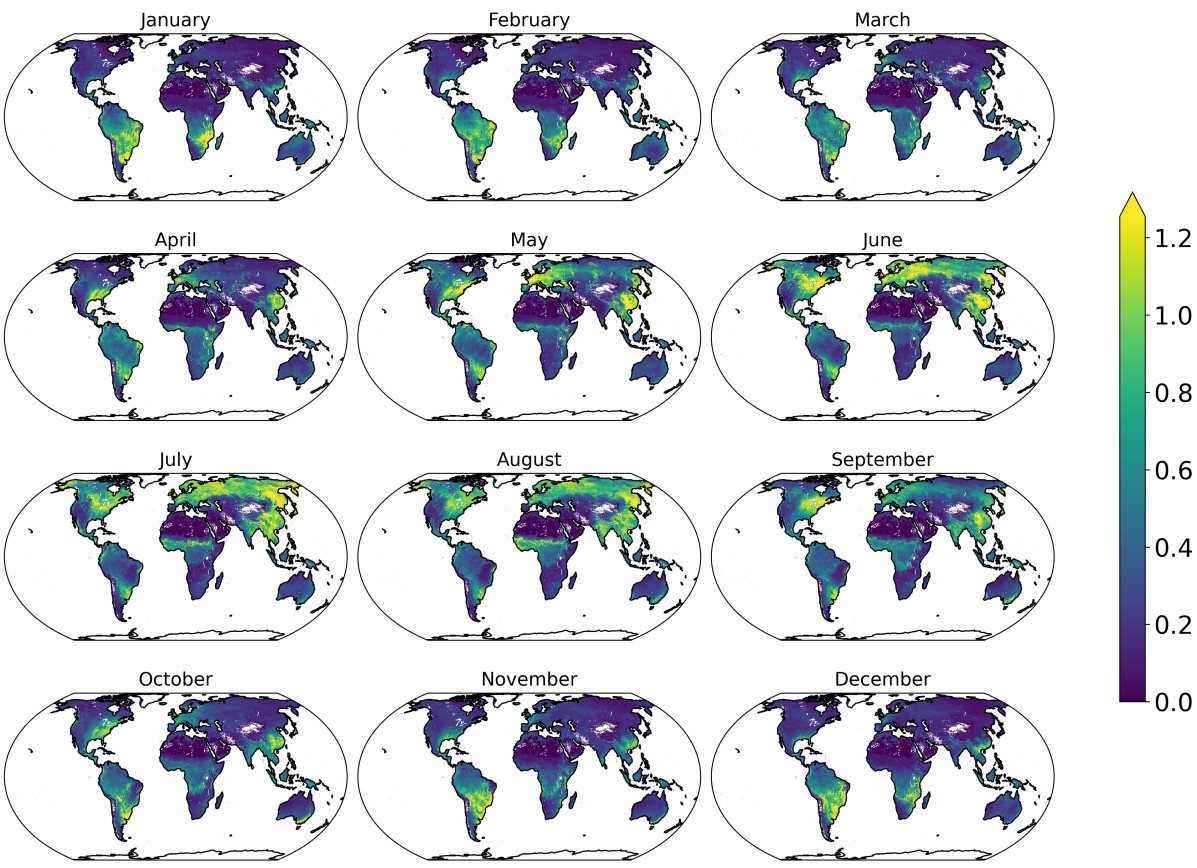

**Figure B1.** EC-STILT monthly mean IAV of NEE, calculated as the monthly mean of the day-of-year standard deviation

*Author contributions.* SU, AB, MR, WP designed the study. SU performed the analysis, and drafted the manuscript. AB, MR, WP, SB
provided analysis and support. JN, SW, MJ provided access and support with the FLUXCOM framework, as well as scientific feedback and
support. LH provided the HUN tall-tower observations. All authors revised and edited the text.

*Competing interests.* The authors declare that they have no conflict of interest.

*Acknowledgements.* We would like to thank the FLUXCOM team for their structural support, feedback and discussion. The authors would
like to thank the eddy covariance community, particularly FLUXNET and the associated regional networks, the European Integrated Carbon



Observation System (ICOS) and AmeriFlux. We would like to thank Dr. Jost Lavric for the use of ATTO observational data. We would like to thank Dieu Anh Tran for providing and facilitating our use of the ZOTTO observational data. The Authors would like to thank the producers of the inversion data included in this study. We would like to thank Dr. Saqr Munassar and Dr. Thomas Koch for providing STILT footprint runs for the European domain. This research was funded by the European Research Council (ERC) Synergy Grant 'Understanding and modeling the Earth System with Machine Learning (USMILE)' under the Horizon 2020 research and innovation programme (Grant Agreement No.

855187). SB and the ATTO project were funded by the German Federal Ministry of Education and Research (BMBF, contracts 01LB1001A and 01LK1602A). The ATTO project is furthermore funded by the Brazilian Ministério da Ciência, Tecnologia e Inovação (MCTI/FINEP contract 01.11.01248.00) and the Max Planck Society.



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
