# Peer review of "Constraining a data-driven CO2 flux model by ecosystem and atmospheric observations using atmospheric transport"

_EGUsphere, 2025_

## Author Comment (AC1)

**To the Editor:**

We apologize for taking more time than planned to address the concerns of the reviewers. We note that we have performed substantial additional analyses based on their request, with the goal to elucidate the transfer of information in our machine-learning approach. We have succeeded in this as detailed below, and it is reflected in the revised manuscript we have prepared in the process. Thereto we have added a new section on model evaluation in the Methods section (line 241). In the Results section we have replaced the previous IAV attribution subsection (previous manuscript line 283) with an improved IAV attribution analysis (line 323) and added a new subsection and analysis on learning in feature space (line 360). We then add a Discussion section to integrate the new results into the main argument of the manuscript.

We next provide a point-by-point response to the reviews, with the original remarks from reviewer #1 (RC1) and #2 (RC2) in blue, and our response in regular font.

**Reviewer 1)**

R1C1: The authors should more clearly explain how EC-STILT, using only one year of data from each of the three atmospheric CO2 observation sites, was able not only to correct biases in global NEE but also to more than double the global total NEE interannual variability (IAV) compared to X-BASE. In the annual regional NEE estimates, EC-STILT shows larger deviations from inversion estimates than X-BASE in regions with atmospheric constraints, such as the Eurasian Boreal and South American Tropical regions. Meanwhile, regions without direct atmospheric constraints—such as South American Temperate, Southern Africa, and Tropical Asia—show substantial increases in NEE IAV, contributing significantly to the increase in global IAV. Based on these results, it is difficult to understand how atmospheric constraints led to improved global NEE estimates and their IAV.

Thank you for your thoughtful assessment of our study. We agree that the submitted manuscript did not adequately describe our hypothesis of how our model makes use of limited spatial and temporal observations to improve over the full study time. And we did not provide an adequate analysis to demonstrate our hypothesis. We have removed the section on attribution of IAV based on the annual cycle of IAV, and replaced it with a new analysis (described above under R2C1) which we hope will better address your concerns. We show that when the training sets are compared in feature space, the performance by region is explained by the relationship of the training distribution to the natural distribution. We believe that this confirms our hypothesis and provides a mechanism by which a data-driven model can improve its performance over time without a long-term constraint.

R1C2: The authors state, "This is because EC-STILT learns its land-surface response in environmental space of the features instead of in geographic space like an inversion." If this interpretation is correct, then the neural network within EC-STILT adjusts biome-specific NEE sensitivities to environmental drivers (e.g., temperature or moisture) in a way that minimizes the loss function. For example, the model may predict stronger NEE sensitivity to moisture in

tropical forests, leading to increased IAV in regions with high moisture variability. But does this sensitivity enhancement improve IAV only in some regions within a biome and not others, due to spatial heterogeneity? While the neural network may function as a black box, I believe the authors could still provide further insight based on available model outputs. For example, exploring differences in learned climate/environmental sensitivities of NEE between EC\_STILT and X-BASE by regions and/or biome types could help readers better understand why the model produced the observed results.

We agree that the submitted manuscript does not adequately demonstrate this hypothesis. We believe that our new sections in Results (Sec. 4.5) and Discussion (lines 430) described in R2C1 provide an analysis that can partially describe the mapping from learned distribution in biophysical feature-space to NEE space.

R1C3: It is unclear why the authors chose to use only three tall-tower atmospheric CO2 observations, given the availability of long-term surface, aircraft, and satellite-based datasets. Was there a decrease in model performance when more observations were included? Or was the goal to test the efficiency of the system using a minimal number of atmospheric constraints?

The decision to only include a limited set of towers and years was taken because we faced computational limits or barriers both in dataset creation and training. We hypothesize in the paper that more towers might add information, but with regards to the new section describing the distributional aspect of the learning process. We will add the paragraph above (R2C1) describing the computational bottle-necks that led us to select only three towers over 1 year.

R1C4: The current EC-STILT system shows substantial regional deviations from inversion-based estimates, with higher RMSE than X-BASE in some regions. While inversion estimates are not ground truth, this suggests that the information from just three sites may be insufficient to improve regional NEE distributions. Although the authors mention plans to address this in future work, it would strengthen the manuscript to provide at least a preliminary assessment—such as how results change when incorporating background in-situ measurements from NOAA's ObsPack data.

We have added a paragraph to address this concern (lines 405), which is included below in the response to R2C1.

**Detailed comments**

 Line 2: The phrase "terrestrial land-atmosphere flux of CO2" seems to refer more closely to net land flux or net biosphere exchange rather than net ecosystem exchange (NEE). I suggest using "net ecosystem exchange" to make the intended meaning clearer.

Fixed, thank you.

2. Lines 11–17: As noted earlier, while your study effectively reduces global NEE biases and improves interannual variability using a limited number of atmospheric CO2 observations, it also leads to increased regional biases—assuming that inversion estimates are reasonably close to the truth. Since a broader set of atmospheric CO2 data is available, including surface, aircraft, and satellite observations, it seems likely that incorporating more of them could improve both global and regional NEE estimates. Could you clarify why only a limited set of atmospheric observations was used in this study?

As discussed above, we have added a paragraph which we believe addresses this comment (lines 405), which is included above in the response to R2C1.

3. Lines 62–71: You mention a key limitation of the previous work by Upton et al. (2024)—that the additional atmospheric information was aggregated and provided no added value for resolving the spatial distribution of NEE. How your EC-STILT approach overcome this limitation? Could you explicitly discuss which aspects of EC-STILT (i.e., global and regional NEE estimates and their IAV) show improvement over the previous work, which do not, and what underlying factors might explain these differences.

This is addressed in new text in the discussion (lines 445) which is included in the response to R2C1.

4. Line 125: As you discuss later in the manuscript, inversion-based terrestrial biosphere flux estimates include not only fire emissions but also lateral fluxes. Since your study assumes that NEE corresponds to the inversion estimate with fire emissions removed, it would be helpful to state this assumption clearly at this point in the text.

We include the new text in the Data section (lines 180):

We do not include lateral transport fluxes in the current analysis. Crop and wood harvest are important for regional and long term accounting (Ciais et al., 2022), but are not critical for the instantaneous carbon budget that is represented in Eq. 1. The potential impact of riverine transport are discussed below in Sec. 5.1.

**Line** 176: Please provide more detail on how the lateral boundary conditions for the region are derived from the 3D CO2 fields provided by Jena CarbonScope, and how these are applied in Equation (1).

We have added text, described above (R2 detailed comment 3) which addresses this concern.

5. Lines 205–206: The way uncertainty is defined and prescribed seems to be a critical component of your system, but the explanation provided is not sufficiently detailed. Could you clarify how uncertainties were defined in your framework, and how the relative weighting between atmospheric constraints and eddy-covariance observation constraints was determined?

The relative weight of the two terms is a learned value in our model, as described at the end of section 3.4. This method allows the model to estimate an optimal mix of the terms based on a learned estimate of homoscedastic uncertainty for each of the different terms of the objective function and is dependent on the inherent noise in the data, rather than the scale or quality of the inputs. We added the text described in R2C4 to better describe this process.

We use our cross-fold validation members to represent the model uncertainty in the study, presenting the member mean and standard deviation. We intended to provide a proof-of-concept, evaluating the value of including atmospheric mole fraction observations in data-driven estimates of NEE. Nevertheless, to address the sources of this model uncertainty, In line with R2C1, we now present a mechanism (Fig 9, section 4.5) which we believe adds nuance to the existing discussion of model uncertainty.

6. Lines 233–234: The phrase "with only local driver variables, and no atmospheric information" is somewhat unclear. It would be helpful to revise this sentence to more specifically describe what is meant by "local drivers" and "no atmospheric information".

The text (line 271) now reads:

When producing an estimate of global NEE, the model takes the driver variables across the full land surface as inputs, but does not access any STILT footprint data, LBC or NBF data, or mole-fraction data, which are only used in constraint during training.

7. Figure 4: Please consider adding a panel showing the annual mean NEE from X-BASE, so that readers can directly compare it with the EC-STILT results. Additionally, for the panel showing the difference between EC-STILT and X-BASE, it would be helpful either to adjust the colorbar style or to use the same colorbar range as in Figure 4A to facilitate visual comparison.

Figure 4 has been updated to include the X-BASE panel. The color bar for the difference has the original scale, but I believe that the new layout makes the visual interpretation easier. Additionally, the standard deviation panel has been updated. The previous version had a bug in the calculation and has been corrected.

(CAPTION) Figure 4. Spatial distribution of global annual NEE A) Mean annual NEE for EC-STILT in gC m-2 day-1 B) Mean annual NEE for X-BASE in gC m-2 day-1 C) The difference in mean annual NEE between EC-STILT and X-BASE in gC m-2 day-1. D) The standard deviation of mean annual NEE for the EC-STILT 10-member ensemble in gC m-2 day-1

8. **Table 1 and Figure 5**: Some values in the text, table, and Figure 5 are inconsistent. Also, bold formatting in Figure 5 seems to incorrectly indicate better performance in some cases—for example, the annual RMSE for Australia. Please review and correct these issues.

Thank you, it is now fixed.

9. Lines 291–292: The statement "EC-STILT has modified its response by biome" should be supported by a clearer explanation in the Methods section. Does this mean that the relationship between driver variables and NEE is trained and applied in a biome-specific manner? If so, please clarify how this is implemented.

This section has been replaced by a stronger attribution of IAV and description of the mechanism

10. Lines 292–293: The sentence "When IAV is broken down by month (Fig. 7) across boreal and temperate regions, the increases in monthly IAV occurs during the growing season while in tropical regions show a change during the dry season" is interesting. Could you provide a potential explanation for this pattern, or at least discuss possible mechanisms that might be driving these seasonal differences across regions?

This section has been removed in favor of our new attribution of IAV and demonstration of the learning mechanism described in R2C1 above.

11. Lines 300–303: Could you clarify which figure or table supports this part of the text?

We have created this table in the appendix at the request of the reviewer.

**12. Figure 8**: Some of the values shown in the figure do not match those reported in the main text. Please review and correct these inconsistencies.

Corrected, thank you.

**13. Line 330**: Since the concept of "environmental response" is a central and novel aspect of your system, it would be helpful to provide a more detailed explanation of how this is defined and implemented in the Methods section. This would help readers better interpret your results.

Thank you for pointing out that this was not sufficiently clear. We have added text to the Methods (section 3.1, lines 155, new text underlined), which we hope, along with the new discussion above in R2C1, makes this concept clear to the reader.

These latent spaces allow the model to learn complex non-linear relationships between driver variables and NEE. Because these relationships are discovered in these abstract non-geographic spaces, they can represent complex space-for-time and time-for-space substitutions. We refer to the process of learning these non-linear, non-geographic mappings between the input distribution and NEE as learning in 'environmental response' space.

**Lines 345–347**: Table 1 suggests that the limited atmospheric observations used in your study are not sufficient to effectively embed the atmospheric signal into the EC-STILT ecosystem response or to guide the spatial and temporal distributions of the estimated NEE.

Yes, this phrase is too enthusiastic, and we add 'partially' which we believe is supported by the results.

14. Lines 379–381: It would be helpful to include supporting results for this statement maybe in the supplementary material.

An example of the training results for the three towers used in the STILT operator for a cross-validation member is now included in the appendix.

(CAPTION) Figure A4. Per-tower representative STILT operator training performance (estimated  $\triangle PPM$  CO2 and observed  $\triangle PPM$  CO2) for the final epoch of one ensemble member. The black, dotted line is the one-to-one line. The blue line is the regression line. The mean absolute error (MAE) and Pearson's R are reported for each tower.

**Reviewer 2)**

R2C1: The authors claim that the inclusion of atmospheric CO2 observations from three sites in their model design improved CO2 flux estimates in terms of capturing inter-annual variability (IAV). Due to the unavailability of observational estimates, they compared their outputs with an ensemble of inversion results and reported that the IAV values for those two are closer compared to X-BASE simulated values. But the authors did not explain how atmospheric CO2 observations covering only one year from these three sites add IAV to the model outputs? Further, it is not clear why they did not use other atmospheric CO2 observations (from additional sites) and/or observations covering a longer period than just one year. It is not clear what are the limiting factors for using more atmospheric observations. At this stage, it is not convincing that these limited data can provide inter-annual constraints for simulated outputs.

We thank the reviewer for their critique of our paper. We recognize that the manuscript did not fully demonstrate the mechanism by which IAV is improved given the limited possible direct constraint from a single year of data from a limited number of atmospheric tall tower locations. We have performed substantial new analyses and we can now also provide a more formal view of our understanding of the underlying model mechanism that allows it to better represent IAV.

With regard to the number of towers, this study aimed to provide a proof-of-concept system, so that we limited the number of towers for computational reasons. This is mostly due to the large time required to run STILT for new temporal and spatial domains and to train the machine learning model with an increasing number of atmospheric time-series. We explicitly chose three towers to cover the climate and ecological space as broadly as possible with a limited set of sites, and chose a full year to encompass the full set of ecosystem flux responses within that space. We agree that more towers and more years are very likely to improve the system, but the additional towers and years would be of most benefit if they cover environmental conditions that are not well represented by the current tower set. And although they might yield better fluxes, they would not demonstrate the Method in a substantially different way than we can with the current three towers.

A new paragraph (line 405) is now included in the discussion section to address this point:

We selected a limited subset of towers and training years to balance computational cost and model performance in this proof-of-concept system. The computational cost of additional towers and years is primarily associated with creating STILT footprints for a new temporal and spatial domain, and is costly both in terms of computational and human effort. We selected three sites with the specific goal to achieve good coverage of different geographic, climate and ecological zones. This precluded the use of other towers which we believe might have provided valuable constraints in training. The NOAA ObsPack data (Masarie et al., 2014) could provide a large volume of additional training data, both towers and years. Following the analysis of section 4.5 we find that given the structure of our model, one might be able to increase the performance of the model with a targeted selection of towers. These would optimally represent the natural distribution of the land surface, rather than including all measurement towers across observational networks. Future experiments, similar to the EC representation analysis in (Pallandt et al., 2022), can be performed to identify additional towers or years that might yield largest improvements in our predictions.

In regard to the second part of this comment, we propose to improve the discussion about the information gain between EC-STILT and X-BASE, and how that gain can improve the representation of long-term global patterns such as IAV. We must first consider how these global patterns in NEE, which would seem to depend on learning from a long-term atmospheric signal, emerge from the local, hourly fluxes produced by EC-STILT. The following text, plus figures is included in the revised manuscript:

In the Methods (starting line 241):

We perform several analyses by KG geophysical region to understand how model performance is modified by the atmospheric information available during training. To discover the regions which contribute to the global IAV in EC-STILT, we use the covariance method described in Lee et al. (2023). This method uses the row sum of the covariance matrix, scaled by the sum of the full covariance matrix to estimate the

per-pixel contribution to the IAV, which is in turn additive. We then sum by KG region to determine the relative contribution.

To test our hypothesis that adding an atmospheric constraint improves the representation of the land surface, we estimate the probability distribution functions (PDF) of the model's training data (such as temperature and VPD) by KG region. We also estimate the PDF of the full land surface, as represented by the full dataset used to create a global multi-year estimate of NEE. For the training data we estimate two different PDFs: one PDF of the data provided from the eddy-covariance towers, which is the training set for X-BASE, and one PDF representing additionally the areas of the land-surface which are under the STILT footprints, which is the training set of our model. The footprint data is weighted by the number of times that time and location is used during training. Global and regional PDFs are generated from a random subsample of the full dataset. The subsamples are per-year time series for ten percent of randomly selected spatial locations. To quantify the relationship between the training PDFs and the full PDF, we use two metrics that describe the distance between two distributions; Kullback-Leibler and Jennsson-Shannon.

To understand the impact of the STILT data on our results, we train nearly identical models with the same architecture and random state. We train an EC-only version of our model which lacks the STILT operator, and so has no additional atmospheric information during training. To test for potential influences from the atmospheric inversion data which we use to calculate the LBC timeseries for each tall tower, we also train a model which replaces the hourly LBC with the annual mean LBC for each time step. In this way this mean-LBC model has no time-varying information on the background state of the atmosphere.

**In the Results (starting line 323):**

We find that EC-STILT attributes IAV to tropical drylands in regions Aw (tropical savanah) and Bsh (semi-arid) (Fig. 7). This contrasts with the X-BASE NEE, which attributes IAV to regions Dfb (humid continental), and Dfc (sub-arctic). Both figures show the regional contribution of IAV, relative to the overall IAV of the dataset. This metric captures only the magnitude of the contribution, not the relative accuracy of the inferred IAV, which varies strongly by region (Fig. 5). Recent studies (Metz et al., 2023, 2025; Ahlström et al., 2015; Poulter et al., 2014) also suggest that arid regions are the dominant source of the IAV in global NEE. The constraints that drive this result in our system must come from atmospheric CO2 data, and we will try to trace the source of this information in section 4.5.

(CAPTION) Figure 7. Attribution of IAV by Koppen-Geiger land cover class. The figure on the left is EC-STILT, the figure on right is X-BASE. The value represents the relative contribution of the region to the overall global IAV. The attribution is calculated by the covariance method described in (Lee et al., 2023) using annual detrended NEE anomalies.

**In the Results (starting line 360):**

We analyse in Fig. 9 the coverage of the EC-only and EC-STILT training data in terms of climate and ecological space by approximating the probability density function (PDF) of the two joint distributions in feature-space (TA/VPD, and EVI/NDWI). We compare the relative difference in the Kullback-Leibler distance (KLD) and Jennsen-Shannon metric (JS) between the PDF of the full, or natural distribution of a random subset of all pixels, either globally or by KG class, and the two training sets (Fig,9, Tabs. B1- B3). The two variable pairs were chosen to create easily interpretable visualizations. We use subsets of the full 10-dimensional distribution to save on computational costs.

At the global scale, the distribution of the EC training set is more concentrated in cooler, and moderately productive regions than the EC-STILT set (Fig. 9, top panel, compare the higher density in the warmer, more productive regions). In the second row, we can see specifically how the training data changes the representation of a region which is not directly observed. As with the global distributions, the EC-STILT distribution covers the warmer, more water-stressed regions (Fig. 9, middle panel TA/VPD) and in the wetter, more highly vegetated regions (Fig. 9, middle panel EVI/NDWI).

This distributional approach also explains why the model under-performs in regions that are directly constrained by the atmospheric towers. In the third row, the Köppen-Geiger class Cfb (temperate oceanic) which covers most of western Europe, we can see that in both variable sets, the EC-only distribution is closer to the natural distribution (Fig. 9, bottom row). Because of the environmental learning of EC-STILT, this means that the inclusion of atmospheric towers may reduce the model's skill in this region.

The specific impact of individual towers, and their interactions in feature space, are inconsistent across regions, variable pairs and metrics. In tables B1 to B3, we expand the analysis by quantifying the effect of adding or removing individual atmospheric towers from training. The optimal distribution, shown in bold, characterized by the minimum distance between the two PDFs, is enriched by adding 1-3 towers in 49 of 52 regions across both distributions (Tab. B1, B2). However when the full distribution from all towers are considered (B3), the improvement of the tower-enriched distributions is more modest, with only 9 of 26 regions (including global) having improved coverage by the towers in both variable pairs.

(CAPTION) Figure 9. Comparison between training set distributions in two multivariate spaces. Both columns show the difference in distribution between the EC-only training set (left), and the EC+Towers training set (right), and how they compare with the full distribution of the global or regional data in feature space. The left is TA and VPD, the right is EVI and NDWI. The top row is the full global set, the middle row is Köppen-Geiger class Aw ('tropical savanah'), the bottom row is Köppen-Geiger class Cfb ('temperate oceanic'). The contour lines represent the probability distribution function (PDF) of each distribution. Two metrics are calculated for each: Kullback-Leibler (KLD) and Jennsen-Shannon divergence (JS), which are measures of distance between the two PDFs.

**In the Discussion(line 430):**

Because EC-STILT has no direct access to long-term information about the state of the atmosphere, and yet produces an estimate of global annual NEE which is closer to the

atmospheric inversions, there must be some information within the available training data which includes this information. The identically constructed model, trained without the inclusion of the STILT operator (Sec. 3.6) does not produce an increased IAV (see Appendix fig. B2), which removes the influence of the model architecture. We further evaluate the influence on IAV of the LBC (Eq. 1), which is derived from atmospheric inversions. In the results from our mean-LBC model, this change did not reduce the observed IAV (see Appendix fig. B3), which means the LBC is not responsible for the improvements in the IAV in the standard EC-STILT.

To understand the increased IAV in the tropical dry regions (see section 4.3), we hypothesize that our model can better represent the local NEE responses to climate variations in these regions through a more complete representation of the natural distribution of the biophysical drivers of NEE, as shown in Fig. 9. This improvement over the information provided by the EC towers in feature-space explains EC-STILT's improved long-term performance using only a single year of observations. EC-STILT learns to represent NEE using only 10 variables at an hourly time-step. With no static variables, such as latitude or elevation, or PFT information, the model can be considered largely independent of a particular spatial domain. The model only learns to map from a training distribution of drivers to a training distribution of NEE. The quality and completeness of this training distribution determines the model's capacity to capture the global phenomena.

The results from (Upton et al., 2024) also support the distribution hypothesis presented above. In that study the atmospheric constraint uses a limited number of fixed pixel locations to infer regional totals. This created an NEE product which was much closer in magnitude and seasonality to an ensemble of atmospheric inversions, but did not improve the IAV. This version of atmospheric constraint does not fundamentally change the model's available view of the land surface. From this we can see that the inclusion of training data which fully includes the IAV signal, but that does not improve the distributional representation of regions from which the IAV emerges from the local variance, does not improve the model's ability to capture the IAV.

R2C2:It should be noted that the reported correlation values are significant ( $R^2 = \sim 0.4$ ) but rather weak. This needs to be discussed and put into context. Further, the regional comparison shows that regions without atmospheric constraint (e.g., Southern Africa) show higher correlation values than regions with atmospheric constraint (e.g., Europe - HUN). How can this be explained?

For the first point regarding the R2 we add the following text (line 494):

While the absolute increase in the R2 of IAV with regard to the GCB23 inversions is modest (0.42 for EC-STILT, 0.02 for X-BASE), it represents a meaningful increase over previous data-driven flux models (Jung et al., 2019; Nelson et al., 2024). As seen above

in section 4.2 and figure 7, EC-STILT improves the estimation of IAV in regions in the southern hemisphere which are known to contribute to a large fraction of the global IAV. However EC-STILT fails to improve the representation of IAV in the northern hemisphere, or where the EC observational record dominates. Therefore, the modest gain in R2, can be seen as a meaningful gain in the representation of the land surface in regions which are otherwise poorly represented in the EC record.

For the second point, we have addressed this issue in the new sections above.

R2C3: I am also curious about why the performance of EC-STILT with regard to IAV is not improved much in the European region, even if the atmospheric constraint is available. Again, this needs to be explained and discussed in the manuscript.

We addressed this comment in a new figure (Fig. 9) which shows the distributional relationship between the EC-only and EC-tower representations, and we hope better describes the observed relationship between X-BASE and our model.

R2C4: Furthermore, the relative weights ( $\omega_{EC}$  and  $\omega_{ATM}$ ) in the objective function play an important role in constraining the model. For a reader to better understand the constraints, these weights should be provided and explained for each region.

These weights are not calculated by region, but are global values that the model learns during training to mix the two terms. We will add a figure in the Appendix which shows the evolution of the terms during model training, and new text in the methods (line 219, new text underlined):

This uncertainty is dependent on the inherent noise in the training data, rather than the quality or quantity of training data. For EC-STILT these mixing parameters,  $\sigma^2_{EC}$  and  $\sigma^2_{ATM}$ , are added to the processing chain of the model after its initialization. During training, using the normal backpropagation process that uses the chain rule to attribute and update the free parameters of the neural network according to their contribution to the loss value, the model also updates these mixing parameters. For the individual tasks  $L \in [EC, ATM]$ , the  $\sigma^2_L$  parameter is used to create two terms;  $w_L$  (Eq. 6), and  $s_L$  (Eq. 7). These are then used to calculate the effective loss, balanced by the learned uncertainty of the terms (Eq. 8). The evolution of these terms during training is provided in Appendix Fig. A3.

**Detailed comments:**

1. L89: A map of the locations of eddy-covariance towers used for the study may be included, which will provide an understanding of the data—availability around the globe.

Included in the appendix

 L143: Authors did not properly explain what are the driver variables used in their model to estimate CO2 fluxes (NEE). A table of driver variables used in the EC-STILT model and their sources should be included.

Included in the Data section

3. L192: It is not clear how the authors calculate the LBC and ocean components for each CO2 observation. Is it with the help of STILT footprints?

The ocean term is created using STILT footprints directly, and is better described now in the text referenced below in minor comment #6, the LBC uses the STILT particles which are used to create the footprint.

The following sentences are added (lines 183):

The LBC values are calculated for each tall tower for each observation time. We used the ensemble of STILT trajectories released at each data point to obtain the mean ending position (lat, lon) as well as a mean ending height above the ground. Using this information we sample global 3D fields, in this case the optimized CO2 mole fractions from CarboScope, and obtain a mole fraction associated with a measurement at the tower. We acknowledge that this method is sensitive to biases in the global 3D fields, but for example at ATTO a bias-corrected version of the LBC yielded very similar results to the CarboScope LBC, please see Botia et al. (2022) for the full discussion.

4. L193: How is NEELand estimated? Here, it is not clear how the ecosystem-model is designed for points where there are no eddy-covariance observations.

The model uses the 10 biophysical driver variables for every location and time-step. These can be provided by a mix of EC measurements and MODIS observations, in the case of the EC objective function term in training, or from a global datacube from ERA5 and MODIS data for non-EC locations for the STILT footprint term. This is described in the first sentence of section 3.1, the drivers are described in table 1. We have updated the description of the objective functions (sections 3.3, 3.4) to better reflect the terms in figure 1.

 L208: The authors should explain the back-propagation process with more clarity. It is not clear from the description which and how parameters are getting updated. Also, I think Figure 1 can be improved further to incorporate back-propagation in the final stage.

The sentence:

For EC-STILT these mixing parameters,  $\sigma^2_{EC}$  and  $\sigma^2_{ATM}$ , are added to the processing chain of the model after its initialization, and then during training, the

normal backpropagation process that neural networks use to update their parameters, can also update these mixing parameters.

**Is replaced by:**

For EC-STILT these mixing parameters,  $\sigma^2_{\text{EC}}$  and  $\sigma^2_{\text{ATM}}$ , are added to the processing chain of the model after its initialization. During training, using the normal backpropagation process that uses the chain rule to attribute and update the free parameters of the neural network according to their contribution to the loss value, the model also updates these mixing parameters.

**6. L235: How is IAV calculated?**

We chose to use the simple standard deviation of the annual integral of NEE across the study period (2001-2021), aggregated either globally, or by region. This is now included in the text at line 273 (new text underlined).

When the model is run globally for years 2001-2021, the standard deviation of annual NEE (IAV) of the EC-STILT member mean...

7. L442: What are examples of the 'meaningful non-biogenic flux' terms that are not included? And how would these bias the results?

The following text (line 537) is included in the discussion:

Any biogenic or non-biogenic flux terms which are not included, or adequately represented temporally and spatially will bias the atmospheric target that the model is trying to match in training. An example of this would be the non-fire disturbance fluxes, or regrowth after disturbance. Another potentially important term is the instantaneous riverine flux of CO2, coming from lateral riverine transport. These terms could push the relative carbon balance towards CO2 release (disturbances) or towards CO2 assimilation (regrowth), and during training, the model would attempt to match this new local target balance. As seen above, because of the distributional nature of the model's learning and inference, this new local balance could modify the global NEE response.

**Minor comments:**

- 1. L26 and further occurrences: Nelson\* and Walther\* et al., 2024 -> Nelson et al., 2024
- 2. L40 and further occurrences: Walther\* and Besnard\* et al., 2022 -> Walther et al., 2022

This reference style is included at the request of coauthors, and we hope copernicus publications support the acceptance of dual-first author citations.

3. L75: Time-inverse -> Time-Inverted Corrected, thanks

- L114: Integrated Carbon Observatory System -> Integrated Carbon Observation System Corrected
- 5. L126: RANDERSON et al., 2017 -> Randerson et al., 2017 Corrected
- 6. L193: ΔPPMfoot need to be properly defined (preferably using an equation).

The description of this loss function with relation to the equation

$$PPM_{Obs} - PPM_{LBC} - \Delta PPM_{Ocean} = \Delta PPM_{NEE} + \Delta PPM_{NBF}$$

is now:

For each observation, a pre-computed STILT footprint, along with associated NBF, LBC and ocean data are retrieved. The observed mole fraction, LBC and ocean contributions are used to create the left-hand side of eq 1. The ocean term represents the flux from any pixels under the footprint, transported using the footprint into concentration enhancements at the tower. To create the right-hand side of eq. 1, which is the change in  $CO_2$  mole fractions attributable to fluxes within the footprint ( $\Delta PPM_{foot}$ ). The ecosystem-level model is then run for each non-zero location in the STILT footprint, producing an estimate of local NEE. The NEE inferences and NBF values are transported with the footprint into concentration enhancements at the tower. This produces the two terms  $\Delta PPM_{NEE}$ , and  $\Delta PPM_{NBF}$  which are then added element-wise.

- 7. L199: SL (Eq. 6) -> (Eq. 7) Thanks, it is corrected now.
- 8. L245: Fig. 3 C -> Fig. 4 C Corrected
- 9. Table 1, Figures: Please use the same precision as in the description. Corrected
- 10. L333-335: I suggest splitting this sentence into two for better readability. Corrected